# HIDRA3: deep-learning model for multi-point ensemble sea level forecasting in the presence of tide gauge sensor failures

Marko Rus[1,2], Hrvoje Mihanović[4], Matjaž Ličer[1,3,★], and Matej Kristan[2,★]

[1]Slovenian Environment Agency, Office for Meteorology, Hydrology and Oceanography, Ljubljana, Slovenia
[2]Faculty of Computer and Information Science, Visual Cognitive Systems Lab, University of Ljubljana, Ljubljana, Slovenia
[3]National Institute of Biology, Marine Biology Station, Piran, Slovenia
[4]Institute of Oceanography and Fisheries, Split, Croatia
★These authors contributed equally to this research.
**Correspondence:** Marko Rus (marko.rus@fri.uni-lj.si)

**Abstract.** Accurate modeling of sea level and storm surge dynamics with several day-long temporal horizons is essential for effective coastal flood response and the protection of coastal communities and economies. The classical approach to this challenge involves computationally intensive ocean models that typically calculate sea levels relative to the geoid, which must then be correlated with local tide gauge observations of sea surface heights (SSH). A recently proposed deep learning model, HIDRA2, avoids numerical simulations while delivering competitive forecasts. Its forecast accuracy depends on the availability of a sufficiently large history of recorded SSH observations used in training. This makes HIDRA2 less reliable for locations with less abundant SSH training data. Furthermore, since the inference requires immediate past SSH measurements at input, forecasts cannot be made during temporary tide gauge failures. We address the aforementioned issues with a new architecture, HIDRA3, that considers observations from multiple locations, shares the geophysical encoder across the locations, and constructs a joint latent state, which is decoded into forecasts at individual locations. The new architecture brings several benefits: (i) it improves training at locations with scarce historical SSH data, (ii) it enables predictions even at locations with sensor failures, and (iii) it reliably estimates prediction uncertainties. HIDRA3 is evaluated by jointly training on eleven tide gauge locations along the Adriatic. Results show that HIDRA3 outperforms HIDRA2 and the Mediterranean basin NEMO setup of the Copernicus CMEMS service by ∼15% and ∼13% mean absolute error (MAE) reduction at high SSH values, respectively, setting a solid new state-of-the-art. Forecasting skill does not deteriorate even in the case of simultaneous failure of multiple sensors in the basin or when predicting solely from the tide gauges far outside the Rossby radius of a failed sensor. Furthermore, HIDRA3 shows remarkable performance at substantially smaller amounts of training data compared with HIDRA2, making it appropriate for sea level forecasting in basins with large regional variability in the available tide gauge data.

## 1 Introduction

Sea surface height (SSH) forecasting is a critical modeling task for two primary practical reasons. Elevated sea levels can result in severe flooding of densely populated coastal towns, while lower sea levels may restrict marine cargo traffic from navigating

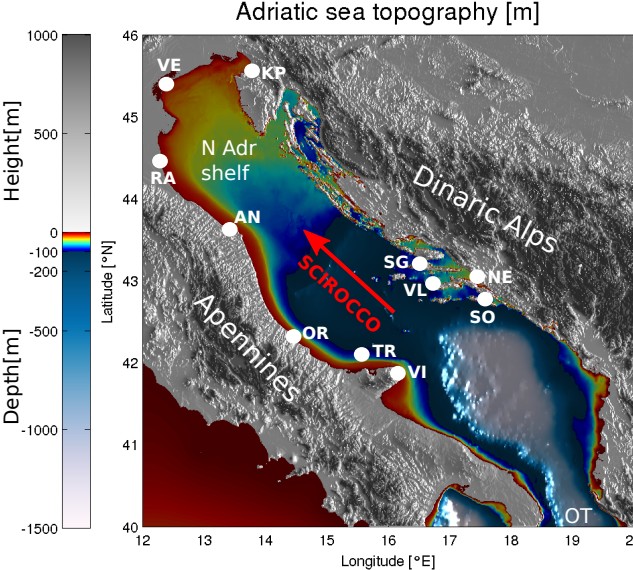

**Figure 1.** Topography and bathymetry of the Adriatic region. White dots denote tide gauges used in this study. Abbreviations used on the map are as follows: KP - Koper, VE - Venice, RA - Ravenna, AN - Ancona, N Adr Shelf - Northern Adriatic Shelf, OR - Ortona, TR - Tremiti, VI - Vieste, SO - Sobra, SG - Stari Grad, VL - Vela Luka, OT - Otranto strait. The direction of Scirocco is marked with the red arrow. The image was created by the authors based on General Bathymetric Chart of the Oceans (GEBCO) bathymetry and elevation data, available at https://download.gebco.net/ (last access: 14 June 2024).

shallow coastal regions like northern Adriatic, where depths often fall below 15 meters (see Fig. 1). In the Adriatic basin, the challenges posed by high and low sea levels are dynamically distinct. Elevated sea levels are associated with intense pressure
lows and strong winds typical of cyclonic activity in the basin, whereas negative sea level extremes arise during periods of spring tides, combined with high atmospheric pressure due to anticyclonic presence.

   Numerical general circulation models are commonly used for SSH prediction and provide basin-scale 4-dimensional simulations of the full sea state (temperatures, salinities, circulation, sea levels), whose evolution is nontrivial and numerically demanding (Umgiesser et al., 2022; Ferrarin et al., 2020; Madec, 2016). Furthermore, sea level information in these models
typically corresponds to a minor part of the total numerical cost of these simulations, which often involve baroclinic and other processes that have very limited relevance for coastal flood predictions. A further complication arises with the fact that semi-enclosed basins exhibit meteorologically induced standing oscillations or seiches which get superimposed on the tidal sea level signal. This leaves the total sea level highly non-linearly dependent on the time lag between peak tide and peak seiche. A possible remedy to this sensitivity is to employ probabilistic or ensemble modeling (Rus et al., 2023; Bernier and Thompson,
2015; Mel and Lionello, 2014) with the hope of efficiently reproducing the envelope of possible sea levels. This approach, however, comes with a high numerical cost, making it unfeasible to run ensemble predictions multiple times a day at national metocean services.

To limit numerical cost, one effective option is to use 2-dimensional barotropic models, which have demonstrated efficacy in sea level and ensemble modeling (Bajo et al., 2023; Ferrarin et al., 2023, 2020). Recently, machine learning has emerged as another promising option for reducing numerical costs while enhancing performance. Early models (Imani et al., 2018) used traditional techniques such as Support Vector Machines (SVMs) (Vapnik, 1999), while Ishida et al. (2020) employed Long Short-Term Memory networks (LSTMs) (Hochreiter and Schmidhuber, 1997) with atmospheric variables for one-hour predictions. Braakmann-Folgmann et al. (2017) extended prediction horizon using a combination of LSTMs and Convolutional Neural Networks (CNNs). Autoregressive neural networks were explored in Hieronymus et al. (2019), increasing temporal resolution and the prediction horizon. While these methods were orders of magnitude faster than their numerical counterparts, their accuracy was generally inferior.

A decisive improvement of accuracy was possible through the employment of deep convolutional architectures from computer vision, such as models from the HIDRA (HIgh-performance Deep tidal Residual estimation method using Atmospheric data) modeling family (Žust et al., 2021; Rus et al., 2023). HIDRA1 and HIDRA2 networks avoid the need for basin-scale simulations of full ocean state and focus the network capacity to only predict sea levels at a single geographic location. HIDRA2 (Rus et al., 2023) has been shown to substantially outperform several numerical models in speed as well as in accuracy of sea level predictions in Koper (northern Adriatic) and is currently implemented also in the Baltic basin on Estonian coasts (Barzandeh et al., 2024).

The HIDRA2 model currently provides SSH forecasts with a 6-day temporal horizon. Its inputs consist of atmospheric winds and pressures as well as SSH observations in the past 24 hours at the prediction location. This means that a sufficiently large history of measured SSH and atmospheric values is required to train HIDRA2 at each geographic location separately. While the atmospheric values are available from sources such as ECMWF (Leutbecher and Palmer, 2007), the SSH values can only be obtained from in-situ tide gauges. This brings two main drawbacks. First, for some locations, a long history of SSH measurements is not available, which limits the training capability of HIDRA2, which has to learn not only the SSH information encoding but also the relevant atmospheric features, the input data fusion process and the prediction heads. Second, in case of sensor failure, immediate past observations are unavailable and predictions cannot be made, several attempts were however made to remedy this situation (Lee and Park, 2016; Vieira et al., 2020).

To address these limitations, we introduce HIDRA3, a new deep-learning architecture designed to simultaneously learn from and predict SSH across multiple tide gauge locations. This new multi-location formulation is the main contribution of this paper. HIDRA3 can further handle missing values in observation datasets at input locations and implicitly reconstruct them in latent space from other available locations. In addition to SSH values, HIDRA3 also predicts SSH forecast uncertainties, which enhances the interpretability of the output.

HIDRA3 thus brings forward several benefits. The prediction accuracy at each individual location is improved by jointly utilizing information from multiple locations. A single HIDRA3 is trained for all locations, thus many parameters are shared between the locations. This means that locations with scarce historical training data benefit from locations with abundant data. Additionally, this enables predictions even during sensor failures, as HIDRA3 exploits all available observed information at

other locations to obtain the predictions at the locations with the missing SSH values. Finally, due to its numerical efficiency, HIDRA3 enables fast and reliable ensemble sea level forecasting multiple times a day.

The remainder of the paper is organized as follows. Section 2 details the new HIDRA3 architecture, Sect. 3 compares
HIDRA3 with state-of-the-art numerical and machine-learning models, while conclusions are drawn in Sect. 4.

## 2 HIDRA3 deep learning sea level model

### 2.1 HIDRA3 training and testing datasets

Our objective is to forecast hourly SSH values for $N = 11$ tide gauges located along the Adriatic coast (Fig. 1) over a three-day period. HIDRA3 achieves this by leveraging a comprehensive set of ocean state parameters. This includes the past 72 hours of
80 available sea level observations from stations shown in Fig. 1, with data availability for each station detailed in Table 1. When calculating the availability, only SSH measurements with 72 preceding measurements available are considered, as required for HIDRA3 input. Besides past SSH measurements, HIDRA3 considers both past and future astronomic tides at the stations, and past and future 72 hours of gridded geophysical variables from atmospheric and ocean numerical models.

| Location | SSH Availability in 2000–2022 | Thresholds [cm] |
|---|---|---|
| Koper | 90.8% | -69.3, 65.7 |
| Venice | 64.6% | -64.3, 61.3 |
| Ancona | 50.4% | -39.9, 44.6 |
| Ortona | 45.3% | -34.0, 39.6 |
| Vieste | 44.9% | -33.3, 36.5 |
| Neretva | 38.9% | -32.6, 37.8 |
| Ravenna | 37.7% | -56.3, 57.2 |
| Sobra | 24.1% | -33.4, 37.0 |
| Stari Grad | 23.9% | -34.0, 38.7 |
| Tremiti | 18.2% | -32.4, 37.0 |
| Vela Luka | 16.6% | -31.9, 38.6 |

Table 1. Availability of SSH measurements between 2000 and 2022 for 11 tide gauge locations used in training and evaluating HIDRA3, and defined thresholds [1st, 99th percentile] for low and high SSH values used in this study. When calculating SSH Availability, only SSH measurements with 72 preceding measurements available are considered, as required for HIDRA3 input. See Fig. 1 for station locations.

The training time window spans two intervals: from 2000 to 2018 and from 2021 to 2022. The testing time window covers
the period from June 2019 to the end of 2020. This specific testing interval was selected due to the occurrence of numerous high SSH and coastal flood events in the northern Adriatic. During training, we assume that the past 72 h of SSH observations are available for at least one tide gauge in the training set, while they may be missing at several or all other locations. Note that

very different amounts of tide gauge data are available at different stations; particularly, all stations situated south of Ancona have less than 50% data availability throughout the training interval (refer to Table 1).

In order to eliminate SSH measurement errors, we apply three filters to address three types of errors: (1) sensor freeze, which results in the same value being produced for an extended period of time; (2) extreme outliers; and (3) extreme jumps in the signals. For case (1), we define a threshold of 5 repetitions, as repetitions typically span more than 5 time points. For case (2), we define a location-specific threshold equal to 10 times the standard deviation of measurements and remove all points that are further away from the mean value than that threshold. Note that the threshold needs to be recalculated after each measurement

is removed. For case (3), we examine the first derivative of the signals and define a location-specific threshold equal to 10 times the standard deviation of all first derivatives for some location. When two derivatives with opposite signs are close together (less than 10 time points), the region between them is removed. Again, it is necessary to recalculate the threshold after each removal. The validity of all thresholds was visually verified.

Astronomic tides were calculated from tide gauge data in one-year intervals using the UTIDE Tidal Analysis package for

Python (Codiga, 2011). Using one year intervals for tidal analysis are common approach to compensate for the unresolved low frequency signals in the tidal signal, like the 18.6 year oscillation due to the precession of lunar nodes (Pawlowicz et al., 2002). For each one-year interval, tidal constituents are inferred from the past year of observations to ensure that no information beyond the prediction point is used. The only exception is in the first year of measurements for each location, when the first year of measurements is used to calculate the first year of tides. This approach is beneficial for training the model and does not

affect the evaluation data, as all tide gauges have measurements spanning at least one year prior to the test period.

In the context of this analysis, a dataset of sea-level extremes was constructed for each station. SSH readings are categorized as *low* if they are below the 1st percentile, and as *high* if they surpass the 99th percentile of the observed values at the respective location. During evaluation (Sect. 3.2), metrics are calculated for all SSH values, as well as for low and high values. This helps assess the model's performance in predicting both tails of extreme SSH values: high values are relevant for coastal flood

warnings while low values restrict marine traffic in the shallow north of the basin. The thresholds determined are listed in Table 1.

Tide gauges in the northern Adriatic (Koper, Venice, Ravenna) and those in the middle and south Adriatic (Ancona, Ortona, Tremiti, Vieste, Sobra, Vela Luka, Neretva, Stari Grad) form two separate clusters that fall within the barotropic Rossby radius of each other and thus exhibit similar SSH phases of high and low sea levels. This is illustrated in Fig. 2, which depicts the

SSH at all stations, and in Fig. 3, which shows mean absolute differences between all stations. These mean absolute differences were calculated using overlapping data from each pair of locations, and they can be interpreted as estimates of the increase in mean absolute error (MAE) when applying some model's forecast from one location to another.

ERA5 reanalysis data (Hersbach et al., 2018) was employed for training purposes, while ECMWF Ensemble Prediction System (EPS) data (Leutbecher and Palmer, 2007) was employed for evaluation to reflect the practical forecasting setup in

which future reanalysis does not exist and forecasts are used. To independently verify the impact of the training set, HIDRA3 was also retrained on ECMWF EPS data instead of ERA5. When evaluated on the ECMWF EPS dataset, this led to a slight increase in MAE (1.7%). We consequently decided to select the ERA5 dataset for the training.

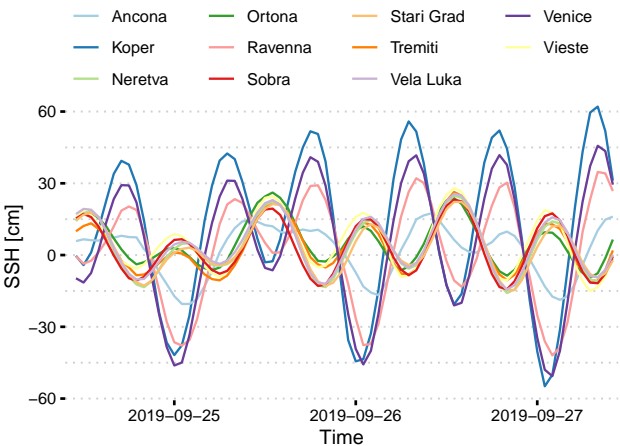

**Figure 2.** A representative example period of SSH from all 11 tide gauges used in training and evaluation of HIDRA3.

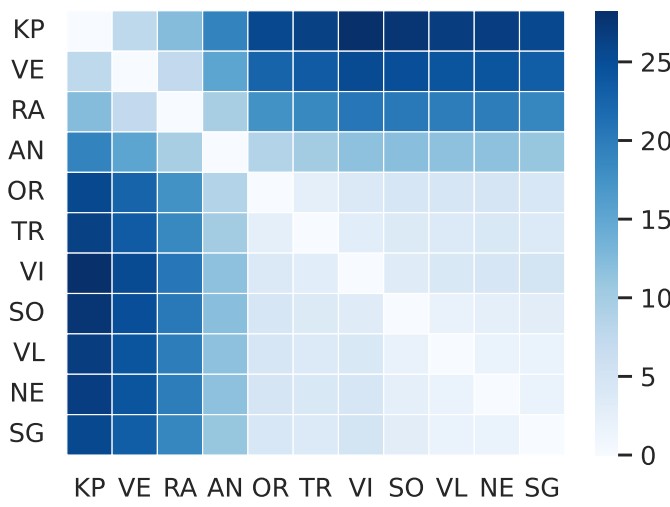

**Figure 3.** Mean absolute differences [cm] of SSH measurements between different tide gauge locations. These differences estimate the increase in MAE when applying some model's forecast from one location to another. Abbreviations used here are: KP - Koper, VE - Venice, RA - Ravenna, AN - Ancona, OR - Ortona, TR - Tremiti, VI - Vieste, SO - Sobra, VL - Vela Luka, NE - Neretva, and SG - Stari Grad.

The metocean model training dataset used in this study consists of 10-meter winds, mean sea level air pressure, sea surface temperature (SST), mean wave direction, mean wave period and significant height of combined wind waves and swell. All atmospheric model input fields were spatially cropped to the Adriatic basin and subsampled to a $9 \times 12$ spatial grid, following our previous work (Rus et al., 2023).

## 2.2 HIDRA3 model architecture

HIDRA3 proceeds as follows (see Fig. 4 for the architecture). Geophysical variables (wind, air pressure, SST, waves) are en-
coded once for all locations in the Geophysical encoder module (Sect. 2.2.1), while the sea level data (full SSH and astronomic
tide) and geophysical features are processed separately for each location in the Feature extraction module (Sect. 2.2.2). Fea-
tures are fused in the Feature fusion module (Sect. 2.2.3), which takes sea-level features from locations with available data and
encodes them into a single joint state feature vector $\mathbf{s}$, which is decoded by the SSH regression head (Sect. 2.2.4) into the SSH
predictions at the individual geographic locations. Finally, the Uncertainty estimation module predicts forecast uncertainties
(Sect. 2.2.5).

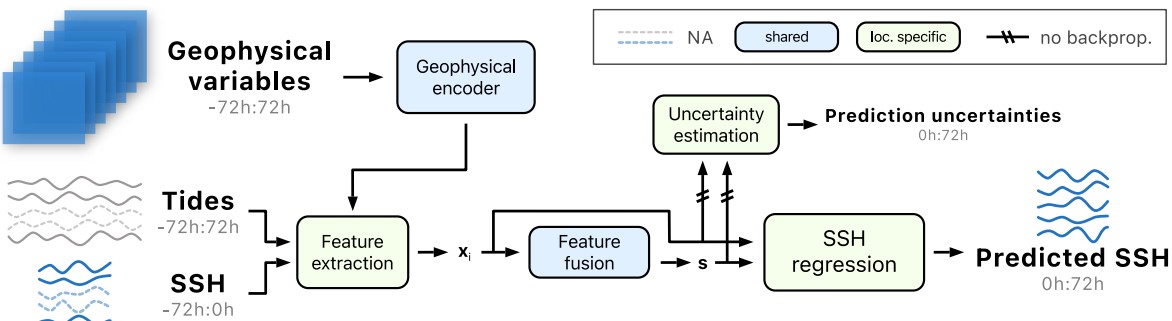

**Figure 4.** The HIDRA3 architecture. The Geophysical encoder encodes pressure, wind, SST and waves. The Feature extraction module
extracts features separately for each location, creating feature vectors $\mathbf{x}_i$, which are then fused in the Feature fusion module. The SSH
regression module produces SSH forecasts, while the Uncertainty estimation module predicts their standard deviations. The location-specific
modules and the modules shared among the locations are depicted in green and blue, respectively. Missing SSH data is denoted by a dashed
curve. The notation $a$:$b$ indicates hourly data points from the interval $(a, b]$, while the prediction point is at the index 0.

### 2.2.1 Geophysical encoder module

The Geophysical encoder takes past and future, i.e. -72:72 h in hourly steps, of geophysical variables as the input: wind ($\mathbf{v} =$
$(u_{10}, v_{10}) \in \mathbb{R}^{2 \times W \times H}$), pressure ($\mathbf{p} \in \mathbb{R}^{W \times H}$), sea surface temperature ($\mathbf{T} \in \mathbb{R}^{W \times H}$), and waves ($\mathbf{w} \in \mathbb{R}^{4 \times W \times H}$), where
$W = 9$ and $H = 12$ are the spatial dimensions of the resampled input fields. The waves tensor $\mathbf{w}$ is composed of the following
four components (hence the dimension $4 \times W \times H$): the first two are sine and cosine encodings of the mean wave direction,
while the remaining two components are the mean wave period and significant height of combined wind waves and swell.

The input variables are processed in two steps. In the first step, variables (wind, pressure, SST and waves) are encoded
separately, and then in the second step, encodings from different variables are fused together. We model each variable indepen-
dently in the first step to avoid modeling low-level interactions between different variables, which would increase the number
of parameters. The first step is shown in Fig. 5. The encoder processes variable of size $c \times 144 \times 9 \times 12$ by a 3D convolution
block with 64 $2 \times 3 \times 3$ kernels with stride $2 \times 2 \times 2$ (the first dimension is temporal, while the second two are spatial), followed

by a ReLU, a 3D dropout (Tompson et al., 2014) and another 3D convolution with 512 $2 \times 4 \times 5$ kernels with stride $2 \times 1 \times 1$. Note that the strides and the kernel sizes are adjusted to reduce the dimensions to $512 \times 36 \times 1 \times 1$. This adjustment aims to compress the information into a low number of features, which helps reduce the risk of overfitting in the network and limits the number of output features of the Geophysical encoder. For the sea surface temperature and wave variables, the number of kernels in the last convolutional layer is 64 instead of 512, forming the output of size $64 \times 36 \times 1 \times 1$. This modification has led to a marginal enhancement in performance, presumably due to the larger number of features representing wind and pressure compared to SST and waves, thereby assigning greater significance to the former variables, which hold more relevance in the context of SSH forecasting.

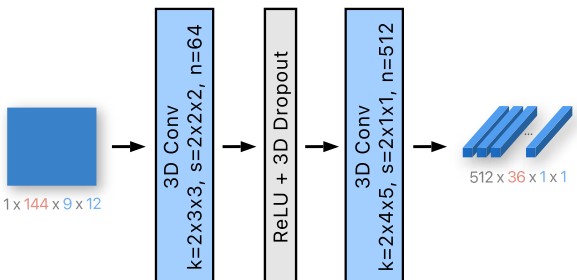

**Figure 5.** The first step of the Geophysical encoder module involves encoding each of the four variables separately. Note that wind has two input channels, while the waves data has four. The encoder consists of two 3D convolutions, reducing spatial dimension to $1 \times 1$. Wind and pressure are encoded into 512 output channels, as depicted in the figure, while sea surface temperature and wave data are encoded into 64 channels. The variables $k$, $s$ and $n$ denote the kernel size, the stride and the number of output channels, respectively. The number of channels is indicated in gray, the size of the temporal dimension is in red, and the spatial dimensions are in blue.

Encodings from all variables are concatenated, resulting in a total of $2 \cdot 512 + 2 \cdot 64 = 1152$ channels. By removing spatial dimensions of size 1, concatenation is of size $1152 \times 36$, which is then in the second step of the Geophysical encoder (see Fig. 6) processed by a 1D convolution with 256 kernels of temporal size 5. We have chosen the kernel size of 5 to increase the receptive field of the layer, which, with this setup, covers approximately 20 hours. We have found this to be effective in preliminary studies (not shown here). Next, two 1D convolutions with 256 kernels of temporal size 1 are applied and followed by a SELU activation (Klambauer et al., 2017), 1D dropout and a residual connection (see Fig. 6). These layers enable the extraction of more complex features from the geophysical variables. After flattening, the output is a single vector of 8192 geophysical features.

### 2.2.2 Feature extraction module

At prediction time, some tide gauges may not have measurements available, e.g., due to temporary sensor failures. Therefore, in the Feature extraction module, the location-specific features $\mathbf{x}_i$ are extracted independently for each location. Then, in the Feature fusion module, the features from all stations are merged, taking into account that the data at some locations is unavailable.

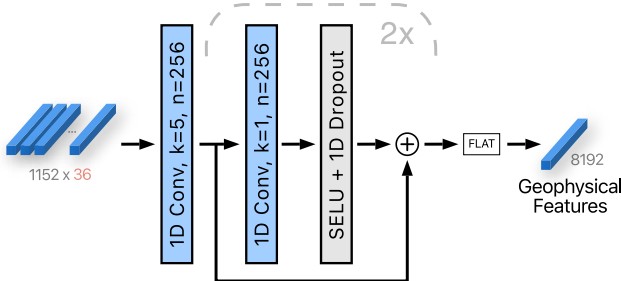

**Figure 6.** The second step of the Geophysical encoder module. Geophysical variables that were encoded independently in the first step are fused to form a single vector. 1D convolutions with residual connections and nonlinearities are applied to produce $256 \times 32$ output, which is then flattened into a single vector of all geophysical features. The variables *k* and *n* denote the kernel size and the number of output channels, respectively. The number of channels is indicated in gray, and the size of the temporal dimension is in red. The block marked with a dashed line is repeated twice.

As is visualized in Fig. 7, The SSH observations (72 values) and the astronomic tide (144 values) are concatenated and encoded with a dense layer (i.e. fully-connected layer) with 512 output channels. Geophysical features are also passed through a dense layer with 512 output channels, whose task is to extract the features relevant to the specific tide gauge. Outputs of
both layers are concatenated and processed with four dense layers with 1024 output channels. A SELU, a dropout and residual connections are applied as shown in Fig. 7. The skip-connection-based approach is chosen for its excellent feature construction capability, which proceeds by adding non-linearly transformed input features as residuals to the original features. This, in practice, leads to better training behavior due to a stronger gradient in the backpropagation phase (He et al., 2016). We chose SELU (Klambauer et al., 2017) for its smoother gradient propagation properties compared to its RELU counterpart, while the
dropout is employed as a classical technique to curb overfitting.

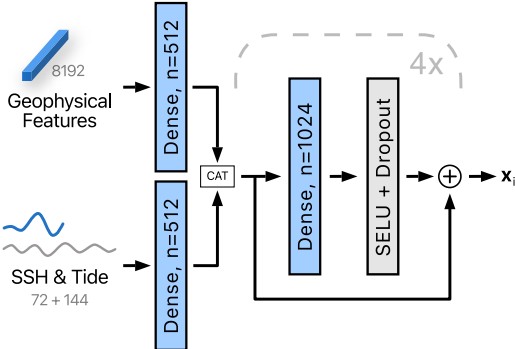

**Figure 7.** The Feature extraction module receives geophysical features, observed SSH measurements and astronomic tide forecast, which are processed independently for each location to extract location-specific features $\mathbf{x}_i$. The variable *n* denotes the number of output channels. The data dimensions are indicated in gray. The block marked with a dashed line is repeated four times.

### 2.2.3 Feature fusion module

As indicated in Fig. 4, the feature fusion module combines the location-specific features $\mathbf{x}_i \in \mathbb{R}^{1024}$, into a joint state vector $\mathbf{s} \in \mathbb{R}^{8192}$. A critical design requirement for the module is robustness to missing data; specifically, the number of location-specific features $\mathbf{x}_i$ may vary, as they are only available for the tide gauges with available measurements.

A partial reconstruction $\mathbf{s}_i$ of the state is computed from each location-specific feature vector $\mathbf{x}_i$ by applying a location-specific dense layer (see Fig. 8). In addition, a weight vector $\mathbf{w}_i$ is computed by applying another location-specific dense layer to $\mathbf{x}_i$. Each coordinate in $\mathbf{w}_i$ reflects the extent by which the particular location contributes to the respective coordinate in the joint state vector. If some tide gauges are non-descriptive, their weights would be reduced during training, lowering their influence on the final state vector $\mathbf{s}$, which is thus computed as

$$\mathbf{s} = \sum_{i \in V} \hat{\mathbf{w}}_i \odot \mathbf{s}_i, \tag{1}$$

where $\odot$ denotes element-wise array multiplication, $\hat{\mathbf{w}}_i$ are the coordinate-normalized weight vectors and $V$ contains indices of tide gauges with available SSH measurements, so that $\mathbf{s}$ is approximated from tide gauges with available measurements. The components $\hat{w}_{i,j}$ are computed by a softmax, i.e.,

$$\hat{w}_{i,j} = \frac{e^{w_{i,j}}}{\sum_{k \in V} e^{w_{k,j}}}, \tag{2}$$

where the softmax ensures that the coordinate-wise weights sum to one across all locations with available measurements.

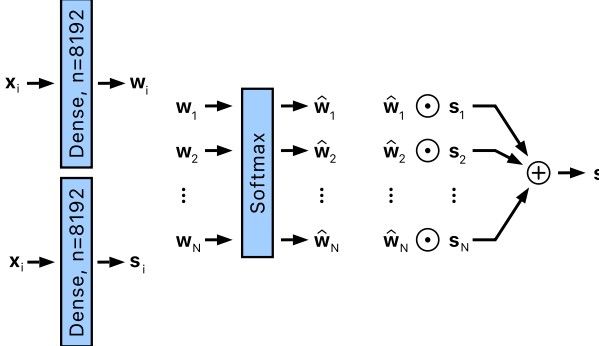

**Figure 8.** The Feature Fusion Module takes features $\mathbf{x}_i$ from location $i$ and uses them to generate weights $\mathbf{w}_i$ and a partial reconstruction of state $\mathbf{s}_i$. These weights and partial reconstructions are combined into a joint state vector $\mathbf{s}$ using weighting and aggregation mechanisms. Locations without available measurements are excluded from the softmax calculation and aggregation. The parameters of the dense layers are specific to each location. The element-wise multiplication is denoted by the symbol $\odot$.

### 2.2.4 SSH regression module

The $i$-th SSH regression module receives the joint state vector $\mathbf{s}$. But since some stations may be deprioritized in the state vector by the Feature fusion module, i.e., due to a smaller amount of training data available for that station, the state vector is

concatenated with a station-specific feature vector $\hat{\mathbf{x}}_i$. The concatenated feature vector is then processed by a dense layer with

72 output features to regress the 72 hourly SSH predictions $\hat{\boldsymbol{\mu}}_i$.

Note that two situations emerge in the construction of the station-specific feature vector $\hat{\mathbf{x}}_i$, depending on the sensor data availability. If the measurements are available for the $i$-th tide-gauge, the station-specific feature vector is simply obtained by passing the features computed for the respective tide-gauge $\mathbf{x}_i$ (Sect. 2.2.2) through a dense layer with 1024 output features. This nonlinear transformation is applied to enable the network to emphasize the part of the feature vector possibly under-

represented in the joint state vector. However, if the measurements are unavailable, e.g. due to sensor failure, then station-specific feature vector is approximated by transforming the joint state vector, i.e. passing it through a dense layer with 1024 output features (see Fig. 9).

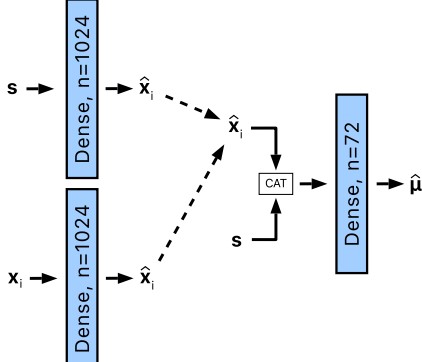

**Figure 9.** The SSH regression module for a location $i$ receives joint state vector $\mathbf{s}$ and location-specific features $\mathbf{x}_i$. The features $\mathbf{x}_i$ are processed by a dense layer to produce the features $\hat{\mathbf{x}}_i$. In cases where measurements from the tide gauge at location $i$ are unavailable, and therefore $\mathbf{x}_i$ is undefined, $\hat{\mathbf{x}}_i$ is approximated from $\mathbf{s}$ using a separate dense layer. The features $\hat{\mathbf{x}}_i$ and $\mathbf{s}$ are then concatenated and passed through a final dense layer to obtain SSH predictions, denoted as $\hat{\boldsymbol{\mu}}_i$.

### 2.2.5   Uncertainty estimation module

Finally, the Uncertainty estimation module predicts the hourly standard deviations corresponding to the SSH estimate $\hat{\boldsymbol{\mu}}_i$.

This layer has the same architecture as the SSH regression module (Sect. 2.2.4) with a slight distinction: it applies a soft-plus function (Glorot et al., 2011) on the output to ensure the positivity of the estimated standard deviation vector $\hat{\boldsymbol{\sigma}}_i$.

### 2.3   Training and implementation details

Training is split into two phases. In the first phase, SSH prediction is trained using the MSE loss between the SSH predictions and the ground truth. We use AdamW (Loshchilov and Hutter, 2017) optimizer with learning rate $1e-5$ and weight decay of

0.001. We apply the cosine annealing (Loshchilov and Hutter, 2016) learning schedule to gradually reduce the learning rate by a factor of 100. To simulate tide gauge failures during training, a random number of tide gauges are turned off with a probability

of 0.5. The batch size is set to 128 data samples, the model is trained for 20 epochs. All input data is standardized by subtracting the mean and dividing by the standard deviation. For each tide gauge location, the mean is calculated independently, while a single standard deviation is computed across all locations. Each geophysical variable is standardized independently of the other variables. Training takes approximately one hour on a computer with an NVIDIA A100 Tensor Core GPU graphics card.

In the second phase, uncertainty prediction is trained. The second phase needs to have a different training dataset, which is necessary to prevent the underestimation of the uncertainty. Hence, only the year 2020 was used to train the second phase. In the second phase, the negative log-likelihood is optimized:

$$\mathcal{L} = -\frac{1}{N} \sum_{i=1}^{N} \sum_{t=1}^{72} \log \frac{1}{\hat{\boldsymbol{\sigma}}_{i,t}} \exp \left[ -\frac{(\boldsymbol{\mu}_{i,t} - \hat{\boldsymbol{\mu}}_{i,t})^2}{\hat{\boldsymbol{\sigma}}_{i,t}^2} \right], \tag{3}$$

where $\boldsymbol{\mu}_{i,t}$ is the ground truth SSH value for the $i$-th tide gauge at prediction lead time $t$, $\hat{\boldsymbol{\mu}}_{i,t}$ is the predicted SSH value, and $\hat{\boldsymbol{\sigma}}_{i,t}$ is the predicted standard deviation. The second phase applies a learning rate of $1\mathrm{e}{-4}$ for 40 epochs; batch size, weight decay, and learning rate schedule are the same as in the first phase.

In operational forecasting and for the evaluation, an ensemble of $n_{\mathrm{ens}} = 50$ atmospheric members is utilized. HIDRA3 processes each member individually, yielding 50 SSH forecasts and associated standard deviations at each time step. These are merged into a single probabilistic forecast through Gaussian moment matching (Kristan and Leonardis, 2013). The prediction is the average of all ensemble forecasts, while the variance is computed as follows:

$$\overline{\sigma}_t^2 = \frac{1}{n_{\mathrm{ens}}} \sum_{i=1}^{n_{\mathrm{ens}}} (\hat{\boldsymbol{\sigma}}_{t,i}^2 + \hat{\boldsymbol{\mu}}_{t,i}^2) - \left( \frac{1}{n_{\mathrm{ens}}} \sum_{i=1}^{n_{\mathrm{ens}}} \hat{\boldsymbol{\mu}}_{t,i} \right)^2, \tag{4}$$

where $\hat{\boldsymbol{\mu}}_{t,i}$ denotes the SSH forecast at time $t$ for ensemble member $i$, and $\hat{\boldsymbol{\sigma}}_{t,i}$ represents the corresponding standard deviation estimate.

### 2.3.1 Network initialization

Considering the complex information flow from many variables that exist on different scales and are represented by latent descriptors of different sizes in the network, care is required at training-time parameters initialization. The weights of convolutional and dense layers are initialized using a standard Xavier initialization (Glorot and Bengio, 2010), while biases are initialized using a normal distribution with a standard deviation of 0.1. In the Feature extraction module, the deeper layers are given lower initial weights, assigning less significance to complex features during the initialization phase. Specifically, for each of the four recurrent dense layers in the Feature extraction module, the weight and bias are scaled by a factor of $0.5^{i-1}$, where $i$ represents the layer's position in the sequence, ranging from 1 to 4.

In the SSH regression module and the Uncertainty estimation module, an additional step of weight scaling is implemented during initialization. In the module $\mathbf{s}$ and $\hat{\mathbf{x}}_i$, which have different dimensions, are concatenated. To ensure that the contributions of $\mathbf{s}$ and $\hat{\mathbf{x}}_i$ to the final output are initially proportionally balanced, the ratio of their sizes is used to scale the weights of the final dense layer in the SSH regression module and the Uncertainty estimation module.

## 2.4 Summary of differences to HIDRA2

HIDRA3 presents a significant advancement over its predecessor, HIDRA2 (Rus et al., 2023), by introducing the capability to simultaneously process data from multiple tide gauge locations. This required a major redesign of the model, as HIDRA3 must effectively handle scenarios where SSH measurements are not available.

The only similar part is the Geophysical encoder module, but with a difference in the way temporal data is processed. In HIDRA2, there is a 4-hour temporal reduction of atmospheric data, while HIDRA3 incorporates the temporal reduction directly into the convolutional operations of the Geophysical encoder module. Additionally, HIDRA3 expands its input data to include sea surface temperature and wave fields, and considers not only the past 24 hours like HIDRA2, but 72 hours before the prediction point.

A notable contribution of HIDRA3 is its capacity for uncertainty quantification, a feature that was absent in HIDRA2. This capability is crucial for assessing the reliability and potential limitations of the SSH forecasts generated by the model.

## 3 Results

### 3.1 NEMO model description

We compare HIDRA3 with the state-of-the-art deep model HIDRA2 (Rus et al., 2023) and with the standard Copernicus Marine Environment Monitoring Service (CMEMS) product MEDSEA_ANALYSISFORECAST_PHY_006_013 (Clementi et al., 2021) numerical model Nucleus for European Modelling of the Ocean (NEMO) v4.2 (Madec, 2016). The Mediterranean Sea Physical Analysis and Forecasting model (MEDSEA_ANALYSISFORECAST_PHY_006_013) operates on a regular grid with a horizontal resolution of 1/24° (approximately 4 km) and 141 vertical levels. It uses a staggered Arakawa C-grid with masking for land areas, and a z* vertical coordinate system with partial cells to accurately represent the model topography. The model incorporates tidal forcing using eight components (M2, S2, N2, K2, K1, O1, P1, Q1) and is forced at its boundaries by the Global analysis and forecast product (GLOBAL_ANALYSISFORECAST_PHY_001_024) on the Atlantic side, while in the Dardanelles Strait, it uses a combination of daily climatological fields from a Marmara Sea model and the global analysis product. Atmospheric forcing comes from ECMWF forecasting product. Initial conditions are taken from the World Ocean Atlas (WOA) 2013 V2 winter climatology as of January 1, 2015. Data assimilation is performed using the OceanVar (3DVAR) scheme, which integrates in-situ vertical profiles of temperature and salinity from ARGO, Glider, and XBT, as well as Sea Level Anomaly (SLA) data from multiple satellites (including Jason 2 & 3, Saral-Altika, Cryosat, Sentinel-3A/3B, Sentinel6A, and HY-2A/B). The hydrodynamic part of the model is coupled to the wave model WaveWatch-III. Further information about the model is available in Clementi et al. (2021).

Since NEMO predicts the full ocean state, including SSH, on a regular grid, the respective tide gauge locations are approximated by the nearest-neighbor locations in the grid. One important thing to note is that ocean models like NEMO calculate sea surface height as a local departure from the geoid in the computational cell. A typical cell dimension is of the order of hundreds of meters. This means that the model's SSH represents a departure from the geoid, averaged over hundreds of squared meters,

and is not directly relatable to the in-situ observations from local tide gauges, which measure local water depth. Therefore, to align NEMO forecasts with local tide gauge water depth, an offset adjustment of the initial 12-hour forecast is necessary to ensure zero bias compared to the respective tide gauge, as explained in Rus et al. (2023).

## 3.2 SSH forecast performance

The following performance measures (Rus et al., 2023) are employed: mean absolute error (MAE), root mean squared error (RMSE), accuracy (ACC), bias, recall (Re), precision (Pr) and F1 score. Additionally, we calculate the normalized mean absolute error (nMAE) by dividing the MAE score for each location by the standard deviation of all historic SSH measurements for that location. These performance metrics are reported in Table 2 separately for all SSH values (*overall*) and for low and high SSH values (see Sect. 2 for the definitions).

| | Model | MAE (cm) | nMAE | RMSE (cm) | ACC (%) | Bias (cm) | Re (%) | Pr (%) | F1 (%) |
|---|---|---|---|---|---|---|---|---|---|
| | NEMO | 2.65 | 0.142 | 3.56 | 97.76 | -0.31 | / | / | / |
| Overall | HIDRA2 | 2.63 | 0.146 | 3.56 | 98.15 | -0.17 | / | / | / |
| | HIDRA3 (ours) | **2.42** | **0.134** | **3.28** | **98.60** | **-0.00** | / | / | / |
| | NEMO | 4.19 | 0.215 | 5.23 | 92.91 | 2.88 | 94.04 | **99.92** | 96.39 |
| Low SSH Values | HIDRA2 | **3.27** | **0.175** | 4.27 | 95.94 | **1.02** | 97.64 | 99.55 | 98.51 |
| | HIDRA3 (ours) | 3.30 | 0.177 | **4.24** | **96.16** | 1.33 | **98.04** | 99.85 | **98.88** |
| | NEMO | 4.68 | 0.244 | 6.19 | 89.14 | -3.02 | 94.53 | **99.40** | 96.79 |
| High SSH Values | HIDRA2 | 4.80 | 0.266 | 6.53 | 89.49 | -2.35 | 96.62 | 97.82 | 97.18 |
| | HIDRA3 (ours) | **4.06** | **0.220** | **5.61** | **91.63** | **-2.06** | **97.58** | 98.67 | **98.09** |

**Table 2.** Performance calculated on all SSH values, low SSH values and high SSH values, averaged over all locations. The proposed HIDRA3 has the best performance overall and on high SSH values, and a comparable performance on low values to HIDRA2.

Table 2 shows that HIDRA3 outperforms both competing models across all metrics computed for all SSH values and for high SSH values. Notably, HIDRA3 achieves an 8.0% lower MAE compared with HIDRA2 (Rus et al., 2023) and an 8.7% lower MAE than NEMO (Madec, 2016). On high SSH values, HIDRA3 achieves a 15.4% lower MAE than HIDRA2 and 13.2% lower MAE than NEMO. These trends are consistent across other evaluation metrics. Furthermore, HIDRA3 demonstrates a near-zero bias and achieves the highest recall on both low and high SSH values among the models. While NEMO has slightly higher precision, this comes at the cost of a lower recall, implying missed events, while HIDRA3 exhibits a solid balance between precision and recall, as evidenced by its highest F1 score both for low and high SSH values.

Figure 10 shows the MAE separately for each tide gauge location. Prediction is most challenging for the northern locations, which yield comparatively larger errors than other locations. This is likely due to shallow waters and topographic amplifications in the northern Adriatic. In these conditions, however, HIDRA3 demonstrates the most significant improvement in MAE when compared with NEMO. In Koper, the MAE is lower by 22.6% and 27.7% on high SSH values, in Venice by 19.0% and

22.2% on high SSH values, and in Ravenna by 13.9% and 30.0% on high SSH values. On the other hand, southern locations show much smaller and comparable MAE errors with both NEMO and HIDRA3, which is also due to overall lower sea level variability in the deeper southern part of the basin with less topographic amplification.

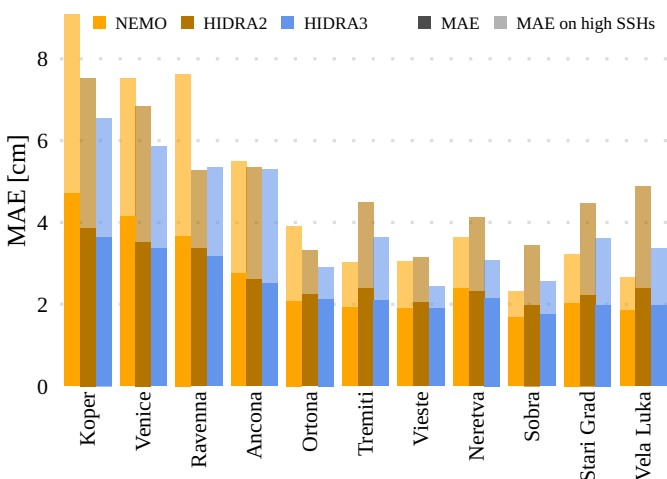

**Figure 10.** The Overall MAE and MAE on high SSH values calculated for different models and tide gauge locations. HIDRA3 significantly outperforms HIDRA2 (Rus et al., 2023), and also outperforms NEMO (Madec, 2016) at northern locations.

To enable a more effective comparison of errors across different locations, we present the normalized MAE scores (nMAE) in Fig. 11. Although the scores are normalized, NEMO still shows the largest errors at northern locations (Koper, Venice and Ravenna). In contrast, HIDRA2 records larger normalized errors at the southern locations, likely due to lower data availability in those areas (see Table 1). HIDRA3 demonstrates the most consistent performance, significantly outperforming NEMO in the northern locations and HIDRA2 in the southern locations. On average, HIDRA3 has a lower nMAE score than both NEMO and HIDRA2 when calculated on all SSH values and high SSH values (see Table 2).

Figure 12 and Fig. 13 visualize December 2019 flooding predictions for Venice and Stari Grad, respectively. The first 24 h of daily predictions were concatenated for each model to get a single time series. In Venice (Fig. 12), HIDRA3 predictions most accurately match the ground truth observations. For instance, NEMO fails to predict the December 14th–16th floods, while HIDRA3 captures them very well. Also, during the peaks in the second part of the month, HIDRA3 is most accurate.

In Stari Grad (Fig. 13), the first peak is missed by all models, while subsequent peaks above the high SSH threshold, as well as low water levels, are predicted most accurately by HIDRA3. NEMO and HIDRA2 often underestimate the range of sea level variability in Stari Grad, with maximums being too low and minimums being too high. HIDRA3, on the other hand, produces a solid forecasting result, even though the training data availability in Stari Grad is merely 23.9% (see Table 1).

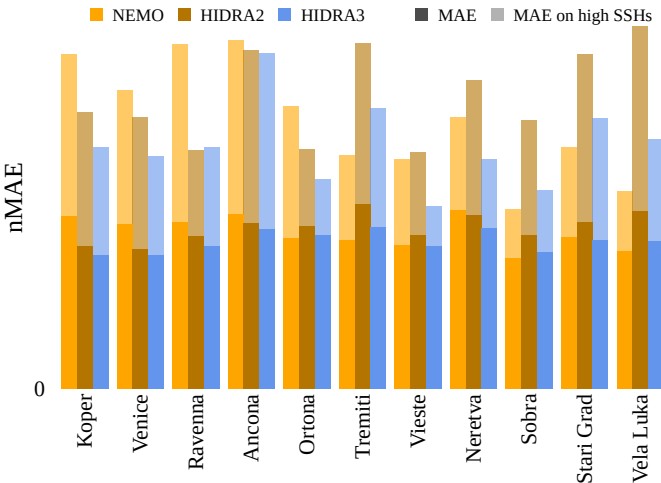

**Figure 11.** The normalized MAE (nMAE) calculated for all SSH values and for high SSH values, across different models and tide gauge locations. HIDRA3 demonstrated the most consistent performance, significantly outperforming NEMO (Madec, 2016) at northern locations (Koper, Venice and Ravenna), and HIDRA2 (Rus et al., 2023) at other locations.

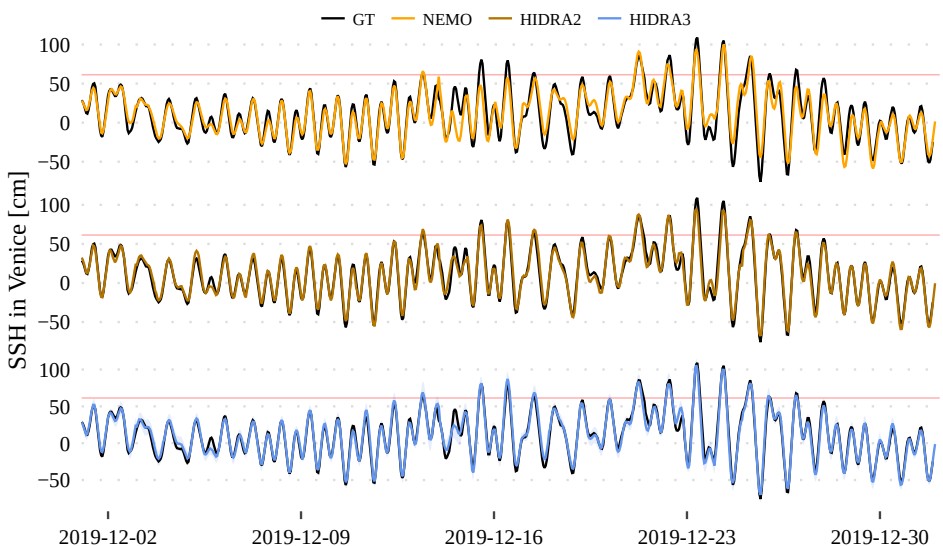

**Figure 12.** Comparison of the HIDRA3, HIDRA2 and NEMO predictions with the ground truth (black line) on the December 2019 floods in Venice. The high SSH threshold is marked with a red line.

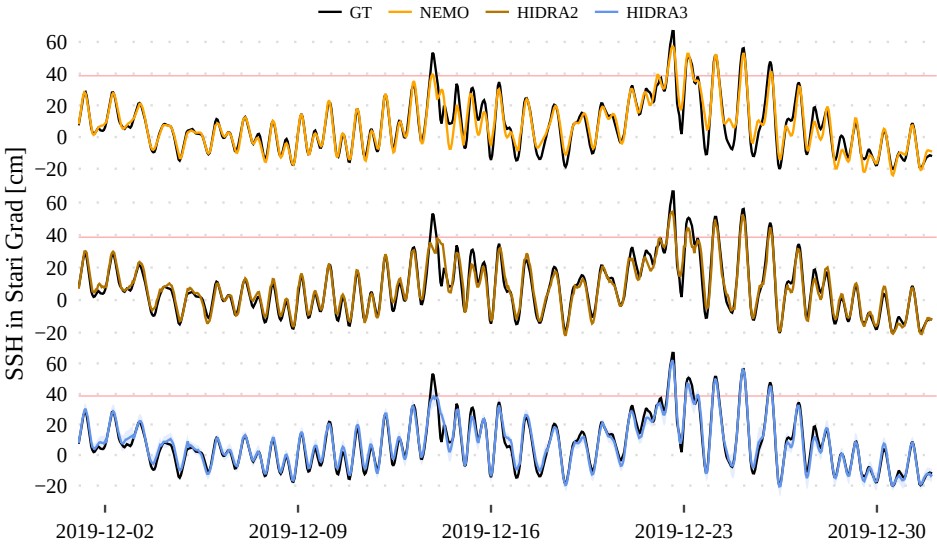

**Figure 13.** Comparison of the HIDRA3, HIDRA2 and NEMO predictions with the ground truth (black line) on the December 2019 floods in Stari Grad, Hvar Island, Croatia. The high SSH threshold is marked with a red line.

### 3.3 Uncertainty estimation analysis

Besides the forecasted SSH values, HIDRA3 also estimates uncertainties by predicting the standard deviation for each forecasted value. To assess the reliability of the estimated standard deviations, we employ the scaled error metric

$$\epsilon_{i,t} = \frac{\boldsymbol{\mu}_{i,t} - \hat{\boldsymbol{\mu}}_{i,t}}{\hat{\boldsymbol{\sigma}}_{i,t}}, \tag{5}$$

as proposed by Barth et al. (2020). This metric quantifies the difference between the ground truth SSH value $\boldsymbol{\mu}_{i,t}$ and the predicted SSH value $\hat{\boldsymbol{\mu}}_{i,t}$, scaled by the estimated standard deviation $\hat{\boldsymbol{\sigma}}_{i,t}$, where $i$ is the tide gauge index and $t$ is the prediction lead time. To assess the accuracy of our predicted uncertainty, we calculate the mean $\mu_\epsilon$ and the standard deviation $\sigma_\epsilon$ of the scaled error. If the predicted and the observed error distributions align, the standard deviation of the scaled error should be $\sigma_\epsilon = 1.0$.

For the second half of 2019 data, which was not used in training, the standard deviation of the scaled error for HIDRA3 is $\sigma_\epsilon = 0.98$. This indicates that HIDRA3 has good uncertainty prediction capabilities, with a slight overestimation of standard deviations of the prediction errors. To further analyze the distribution of the scaled error metric $\epsilon_{i,t}$, we plot its histogram in Fig. 14, along with the ideal model (a zero-centered unit-sigma Gaussian), and with the Gaussian distribution characterized by the estimated mean $\mu_\epsilon$ and standard deviation $\sigma_\epsilon$ of HIDRA3. Note that the estimated Gaussian distribution aligns well

with the distribution of the ideal uncertainty prediction model, suggesting that HIDRA3 has excellent uncertainty prediction capabilities.

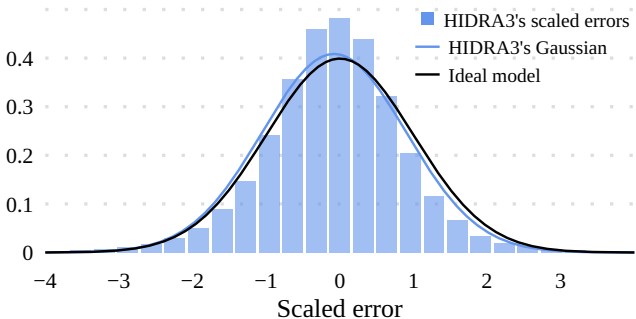

**Figure 14.** Histogram of the scaled error $\epsilon_{i,t}$, overlaid with ideal Gaussian distribution with mean 0 and variance 1, and a Gaussian distribution characterized by the estimated mean $\mu_\epsilon$ and standard deviation $\sigma_\epsilon$ from HIDRA3. The estimated Gaussian distribution aligns well with the ideal one, suggesting excellent uncertainty prediction capabilities of HIDRA3.

## 3.4 Evaluation under tide gauge failures

### 3.4.1 Single tide gauge failure

To evaluate the robustness of HIDRA3 to tide gauge failures, we simulated the failures at each location by making the SSH measurements at a respective location unavailable at prediction time. In such a scenario, HIDRA2 (Rus et al., 2023) is unable to produce predictions. For NEMO, a tide gauge failure primarily means that the bias offset adjustment, which translates predicted SSH (over geoid) into the total water level, is not available. Consequently, we offset the NEMO SSH at the location of the failed tide gauge using the average bias of all the non-failed locations. We refer to this offset version of NEMO results as NEMO$_0$.

The average performance over all locations with the failed tide gauges is shown in Table 3 and visualized in Fig. 15. HIDRA3 achieves a lower MAE than NEMO$_0$ across all locations, most notably in Koper. We emphasize that this is *not* equal to the error of the NEMO model but rather a demonstration of an expected error in case of a sensor failure, where we are forced to infer offset adjustment from available stations.

Note that for the majority of the southern locations, the MAE on high SSH values is higher for HIDRA3 than NEMO$_0$, with the differences being quite small. Nevertheless, in Koper, HIDRA3 outperforms NEMO$_0$ by a large margin and attains a lower mean MAE on high SSH values. This is also reflected in other metrics reported in Table 3.

We next simulated a pair of tide gauge failures: the one close to the river mouth of Neretva River, Croatia, and the Venice tide gauge. Figures 16 and 17 show the corresponding SSH observation time series along with the concatenation of the first 24 h of six daily forecasts. While the previous HIDRA2 cannot predict during tide gauge failures, HIDRA3 seamlessly predicts SSH values with errors lower than NEMO$_0$.

| | Model | MAE (cm) | nMAE | RMSE (cm) | ACC (%) | Bias (cm) | Re (%) | Pr (%) | F1 (%) |
|---|---|---|---|---|---|---|---|---|---|
| Overall | $NEMO_0$ | 3.26 | 0.173 | 4.15 | 95.81 | **0.03** | / | / | / |
| | HIDRA3 (ours) | **2.63** | **0.146** | **3.52** | **98.35** | -0.07 | / | / | / |
| Low SSH | $NEMO_0$ | 4.00 | 0.217 | 4.95 | **96.00** | 3.08 | 97.41 | **99.55** | 98.44 |
| Values | HIDRA3 (ours) | **3.30** | **0.176** | **4.26** | 95.75 | **1.02** | **98.23** | 99.51 | **98.82** |
| High SSH | $NEMO_0$ | 5.12 | 0.255 | 6.48 | 86.82 | -2.96 | 92.20 | **99.81** | 95.24 |
| Values | HIDRA3 (ours) | **4.46** | **0.245** | **6.04** | **89.94** | **-2.32** | **97.38** | 98.81 | **98.06** |

**Table 3.** Performance of HIDRA3 and $NEMO_0$ under the target location tide gauge failure.

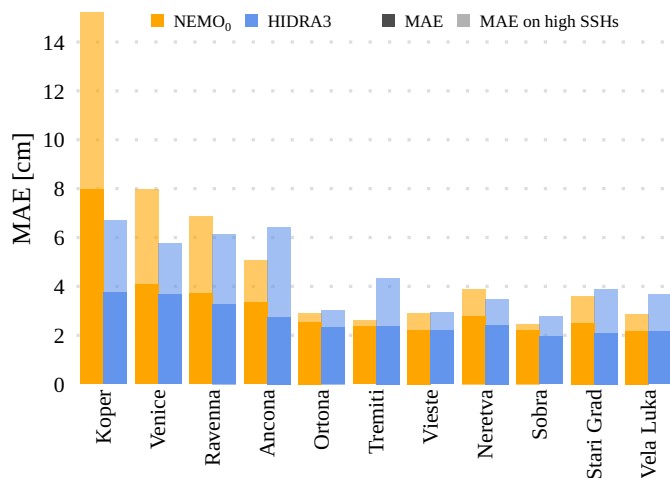

**Figure 15.** The Overall MAE and MAE on high SSH values calculated for locations with simulated single (isolated) sensor failures at stated locations ($x$-axis). Note that the increased $NEMO_0$ errors compared to NEMO errors in Figure 10 are due to a lower bias correction quality during periods of local sensor failures.

### 3.4.2 Extreme scenario of a regional tide gauge failure

To evaluate the model's performance under an extreme scenario of a regional tide gauge failure, we conducted an additional experiment where all northern locations (Koper, Venice, and Ravenna) were disabled. These locations exhibit distinct dynamical characteristics compared to others, as illustrated in Figures 2 and 3. The evaluation of HIDRA3 without SSH measurements from Koper, Venice, and Ravenna is denoted as $HIDRA3_S$. Importantly, the model was not retrained for this experiment; rather, the same model trained on all locations was used, with the specified measurements withheld from the test set. This stress test quantifies HIDRA3 predictions for northern tide gauges based exclusively on data from southern tide gauges.

Figure 18 compares the MAE scores of $HIDRA3_S$, HIDRA3, and NEMO. Surprisingly, $HIDRA3_S$ demonstrates robust forecasting capabilities in the northern locations, despite lacking direct measurements. While the MAE scores are slightly higher

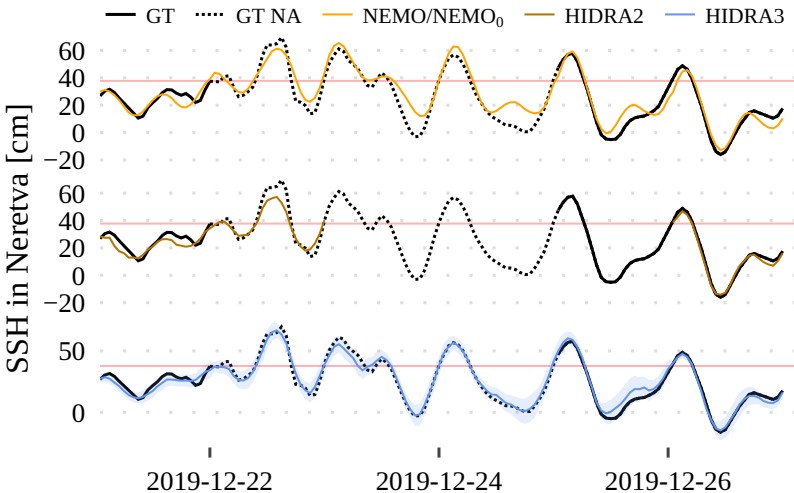

**Figure 16.** Forecasts on December 2019 floods at the Neretva tide gauge with simulated single (isolated) sensor failure. The SSH measurements are shown in black, while the dotted lines indicate the failure simulations. HIDRA2 is unable to make predictions during sensor failure, while HIDRA3 continues to deliver reliable predictions. A two standard deviations band is drawn around HIDRA3 predictions to visualize the estimated uncertainty. Red lines indicate high SSH values.

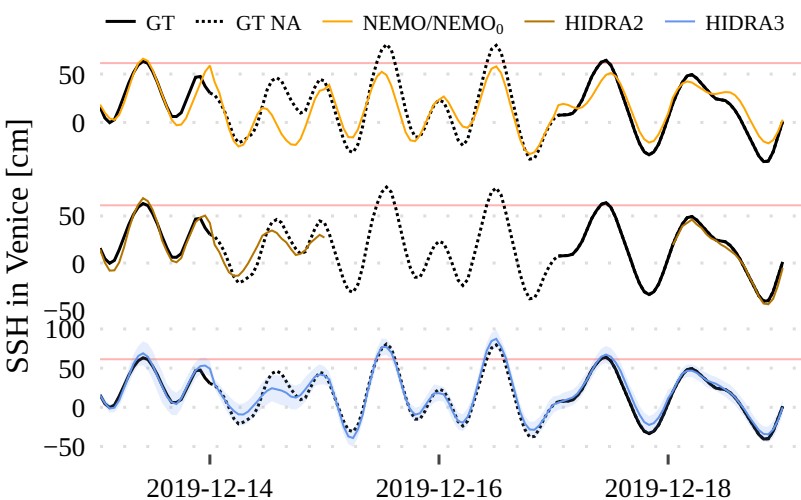

**Figure 17.** Same as Figure 16 but for Venice tide gauge.

than those obtained with all locations as inputs (HIDRA3), they significantly outperform NEMO. In the southern locations, the performance of HIDRA3 and HIDRA3$_S$ is comparable and worse than NEMO. This is expected given the availability of past SSH measurements for these regions.

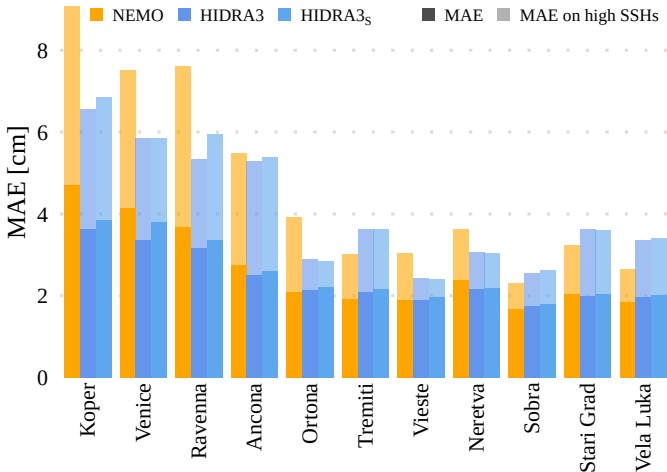

**Figure 18.** Multiple sensor failure scenario. Overall (dark) and high SSH (light) MAE scores for NEMO, HIDRA3, and HIDRA3$_S$. HIDRA3$_S$ represents the evaluation of HIDRA3 with only southern locations as input, excluding sensors in Koper, Venice, and Ravenna.

### 3.5 Influence of the training set size

One of the key hypothesized strengths of HIDRA3 with respect to HIDRA2 is its capability to leverage data from different tide gauges, particularly when SSH data availability is limited and would severely inhibit HIDRA2 training capability. To evaluate this aspect, we re-trained HIDRA3 and HIDRA2 for the Venice location with different historical window sizes while keeping the data from other stations intact. The training set was gradually increased from 1 year worth of data to 10 years worth of training data. The results are shown in Fig. 19, and indicate that the performance of HIDRA2 gradually improves with increasing the training set size and converges if at least 8 years of data are available. On the other hand, HIDRA3 converges rapidly and consistently outperforms HIDRA2 for all considered dataset sizes. Remarkably, even with a single year of training data, HIDRA3 obtains performance comparable to when trained on the entire duration of 10 years.

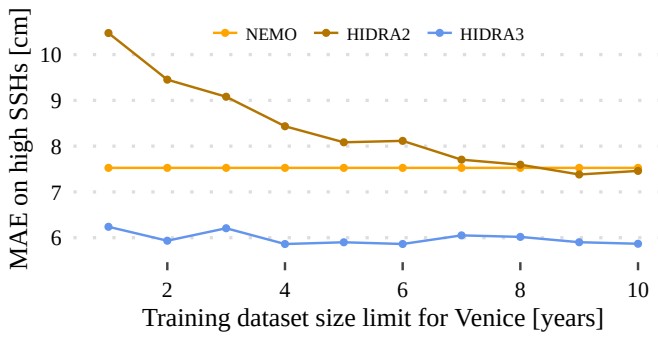

**Figure 19.** MAE on high SSH values when restricting training data in Venice to a maximum of 10 years. HIDRA3 achieves substantially lower MAE than HIDRA2, particularly under limited historical training data.

### 3.6 Ablation study

#### 3.6.1 Importance of considering multiple locations

To further analyze the proposed architecture, we retrained HIDRA3 at each location separately (ignoring all others) to compare it under the same inputs as HIDRA2 – we refer to this variant as $HIDRA3_1$. Results in Table 4 and Fig. 20 show that, on average, $HIDRA3_1$ achieves better performance than both HIDRA2 and NEMO in the overall metrics, with $HIDRA3_1$ particularly excelling on the high SSH values. $HIDRA3_1$ consistently outperforms HIDRA2 across all locations, indicating the superiority of the proposed architecture even when it cannot exploit the information from multiple locations.

Multiple locations nevertheless substantially benefit the training process, as can be seen by comparing $HIDRA3_1$ to HIDRA3. The MAE of $HIDRA3_1$ is higher by 7.4% and 6.9% on high SSH values, meaning that observing multiple locations is beneficial for the overall performance, and that HIDRA3 is able to leverage and combine the information from all stations into a more accurate forecast.

|  | Model | MAE (cm) | nMAE | RMSE (cm) | ACC (%) | Bias (cm) | Re (%) | Pr (%) | F1 (%) |
|---|---|---|---|---|---|---|---|---|---|
| | NEMO | 2.65 | **0.142** | 3.56 | 97.76 | -0.31 | / | / | / |
| Overall | HIDRA2 | 2.63 | 0.146 | 3.56 | 98.15 | -0.17 | / | / | / |
| | $HIDRA3_1$ (ours) | **2.60** | 0.144 | **3.47** | **98.40** | **0.02** | / | / | / |
| | NEMO | 4.19 | 0.215 | 5.23 | 92.91 | 2.88 | 94.04 | **99.92** | 96.39 |
| Low SSH Values | HIDRA2 | **3.27** | **0.175** | **4.27** | **95.94** | **1.02** | **97.64** | 99.55 | **98.51** |
| | $HIDRA3_1$ (ours) | 3.52 | 0.190 | 4.47 | 95.58 | 1.79 | 97.31 | 99.21 | 98.16 |
| | NEMO | 4.68 | 0.244 | 6.19 | 89.14 | -3.02 | 94.53 | **99.40** | 96.79 |
| High SSH Values | HIDRA2 | 4.80 | 0.266 | 6.53 | 89.49 | -2.35 | 96.62 | 97.82 | 97.18 |
| | $HIDRA3_1$ (ours) | **4.34** | **0.239** | **5.98** | **90.94** | **-1.72** | **96.82** | 98.54 | **97.65** |

**Table 4.** Performance of NEMO, HIDRA2 and $HIDRA3_1$, where $HIDRA3_1$ is the model trained separately on every single location.

#### 3.6.2 Impact of sea temperature and waves

To evaluate the importance of using sea surface temperature and wave data to enhance our model, we retrained HIDRA3 by these variables ablated. We first removed only sea surface temperature, which resulted in an average 1.7% increase in the overall MAE and a 1.2% increase in the MAE on high SSHs. Next, we removed only wave data, which led to an average 1.2% increase in the overall MAE and a 2.4% increase in the MAE on high SSHs. Finally, in the third experiment, we removed both sea surface temperature and wave data, which caused an average 0.8% increase in the overall MAE and a 2.7% increase in the MAE on high SSHs. These results suggest that sea surface temperature and wave data are not the main sources of the HIDRA3's excellent performance, however, they do bring a non-negligible contribution and we recommend using them in the SSH prediction tasks.

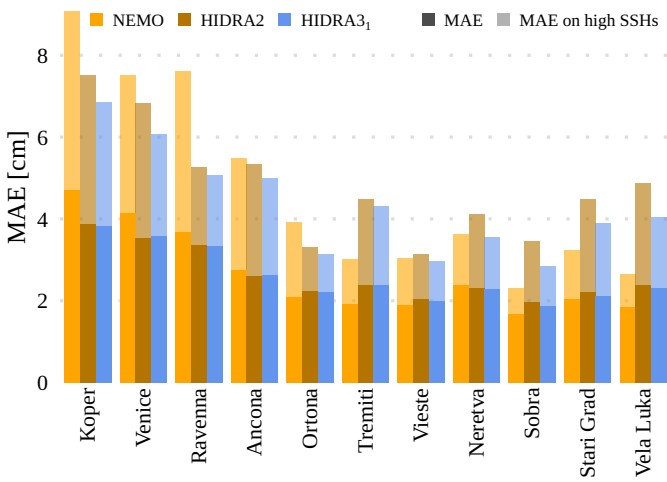

**Figure 20.** MAE comparison between locations, where HIDRA3$_1$ is the model trained separately on every single location.

### 3.6.3 Importance of the skip connection

Finally, we quantify the impact of passing the location-specific features $\mathbf{x}_i$ directly to the SSH regression module in addition to the fused features from other locations, as shown in Fig. 4 in Sect. 2.2.3. For that, we change the SSH regression module so that it only accepts the fused features $\mathbf{s}$ and train the model by limiting training data from Venice to 1 year. This setup was chosen to showcase the performance change in locations with less historical SSH measurements. In Sect. 3.5, we have observed that HIDRA3 achieves MAE on high SSH values of 6.24 cm, but with the changed architecture, MAE rises to 7.25 cm, which is an increase of error by 16.2%. This verifies our architectural design choice, which passes the location-specific features $\mathbf{x}_i$ to the SSH regression module in addition to the joint state, thus improving the location-specific information flow.

### 3.7 HIDRA3 limitations

HIDRA3 can generate forecasts for all locations even if the data from some locations is missing, which is a substantial improvement over its predecessor, HIDRA2. The current implementation of HIDRA3 disregards the entire 72-hour SSH dataset from the previous three days if even a single value of SSH is missing. This might be an overly stringent criterion and we are working to find optimal ways to address this. Furthermore, in a highly unlikely situation when the sensors at *all locations* would fail and the data would be missing for the full 72 h past period, HIDRA3 could not generate forecasts reliably. In this case, a potential solution of using the past HIDRA3 prediction as the input for the current prediction is still applicable. Preliminary simulations of such scenarios indicate that the MAE of HIDRA3 increases by 35% overall and by 43% for high SSH values. Additional work, however, needs to be performed to further constrain HIDRA3 behavior in such scenarios.

## 4 Conclusions

We propose HIDRA3, a new deep-learning architecture for a multi-location sea level prediction with a temporal horizon spanning several days. Machine learning models have recently emerged as a highly competitive alternative to general circulation models, but they are often challenged by the need for extensive periods of training data and operational reliance on real-time SSH observations, which may not always be available due to sensor failures. HIDRA3 addresses these challenges by being robust to individual and multiple tide gauge failures. This approach not only improves training where historical data is scarce, but also enables predictions in the absence of real-time SSH data from a subset of training locations. HIDRA3 furthermore introduces a new reliable module for the estimation of prediction uncertainties, enhancing the interpretability of the forecasts and their integration into downstream operational services.

In a challenging experimental setup, HIDRA3 outperforms both the current state-of-the-art deep learning model HIDRA2 and the CMEMS version of the NEMO general circulation model. Results show excellent prediction capabilities even with limited training data, indicating a remarkable generalization capability of the proposed architecture.

The ability to exploit data from multiple locations for improved individual predictions, robustness to sensor failures, and uncertainty estimation capabilities make HIDRA3 a powerful tool for coastal flood forecasting in regional basins with high variability in the availability of tide gauge data. Our future work will focus on densifying the prediction locations, ultimately leading to fully dense two-dimensional temporal predictions, which will further extend the application outreach of the HIDRA models.

*Code and data availability.* Implementation of HIDRA3 and the code to train and evaluate the model is available in the Git repository https://github.com/rusmarko/HIDRA3 (last access: 27 June 2024). The persistent version of the HIDRA3 source code is available at https://doi.org/10.5281/zenodo.12570449 (Rus et al., 2024a). HIDRA3 pretrained weights, predictions for all 50 ensembles, geophysical training and evaluation data and SSH observations from Koper (Slovenia) are available at https://doi.org/10.5281/zenodo.12571170 (Rus et al., 2024b). Sea level observations from Italian tide gauges are provided by The National Institute for the Environment Protection and Research (ISPRA) and are publicly available at the following address: https://www.mareografico.it (last access: 27 June 2024). Sea level observations from Neretva station are property of Croatian Meteorological and Hydrological Service (DHMZ) and are available upon request at the following address: https://meteo.hr/proizvodi_e.php?section=proizvodi_usluge¶m=services (last access: 27 June 2024). Sea level observations from Sobra, Vela Luka and Stari Grad (Croatia) are provided by the Institute of Oceanography and Fisheries (IOR) and are publicly available at the Intergovernmental Oceanographic Commission Sea Level Station Monitoring Facility (IOC SLSMF; http://www.ioc-sealevelmonitoring.org, last access: 27 June 2024).

*Author contributions.* MR was the main designer of HIDRA3. MK led the machine-learning part of the research and contributed to the design of HIDRA3. ML provided the geophysical background and led the oceanographic part of this research. HM and MR performed quality control over sea level observations. MR, ML and MK wrote the draft of the paper. All authors contributed to the final version of the manuscript.

*Competing interests.* The authors declare that they have no conflict of interest.

*Acknowledgements.* The authors would like to thank the Academic and Research Network of Slovenia - ARNES and the Slovenian National Supercomputing Network - SLING consortium for making the research possible using the ARNES computing cluster. The authors would like to acknowledge the efforts of all the technical staff at the Italian National Institute for the Environment Protection and Research (ISPRA), the Croatian Meteorological and Hydrological Service (DHMZ), The Institute of Oceanography and Fisheries (IOR) and at the Slovenian Environment Agency (ARSO) for maintaining an operational system of tide gauges.

*Financial support.* Matjaž Ličer acknowledges the financial support from the Slovenian Research and Innovation Agency ARIS (contract no. P1-0237). This research was supported in part by ARIS programme P2-0214 and project J2-2506.

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
