# Peer review of "HIDRA3: deep-learning model for multi-point ensemble sea level forecasting in the presence of tide gauge sensor failures"

_EGUsphere, 2024_

## Author Comment (AC1)

Dear editor, dear reviewer.

Before we proceed with point-by-point addressing of the issues raised by the reviewer, we would like to make a very short synopsis of what was done during this revision.

1. Based on the recommendation of Referee #2, we conducted experiments to test the failure of regional tide gauges. In light of this, we propose changing the title of the manuscript by removing the word "robust" and replacing it with a more descriptive phrase "in the presence of tide gauge sensor failures," so the new title reads: "HIDRA3: deep-learning model for multi-point ensemble sea level forecasting in the presence of tide gauge sensor failures."
2. We have added new sections to the manuscript: 2.4 "*Summary of differences to HIDRA2*," 3.1 "*NEMO model description*," 3.3 "*Uncertainty estimation analysis*," 3.4.2 "*Extreme scenario of a regional tide gauge failure*" and 3.7 "*HIDRA3 limitations*".
3. We have provided additional explanations and figures in the architecture description sections, rewriting Section 2.2.3, "*Feature fusion module*," and Section 2.2.4, "*SSH regression module*."
4. We have included explanations for the data quality check procedures and computation of the tidal signal, expanded the analysis of sea surface temperature and waves' impact on the performance of the model, and corrected some typos.

We proceed to a detailed point-by-point response below.

**COMMENT 1:**

**Review of HIDRA3: a robust deep-learning model for multi-point ensemble sea level forecasting.**

**The paper presents a new version of the HIDRA sea level forecasting model. HIDRA3 is a machine learning model with a deep convolutional architecture. The most important update from version 2 is that the current version uses data not just from the local tide gauge it predicts, but also from neighbouring tide gauges, which allows prediction also when the local tide gauge is not operational.**

**The paper is well written, figures are nice and the model architecture and modelling choices are well described. I recommend publications after some minor revisions.**

**RESPONSE 1:**

We thank the reviewer for their encouraging comments.

COMMENT 2:

**General points:**

**1) The manuscript makes the point that their machine learning model outperforms the numerical ocean model NEMO on SSH prediction. But that is not really what is tested. They do get better results on most metrics than what is seen in the specific NEMO run they compare with. However, the performance in that specific NEMO run, says almost nothing about the capabilities of numerical ocean models in general or even of the NEMO modelling systems capabilities. The SSH performance in the NEMO run they compare with depends on modelling choices (resolution, parametrisations, coordinate systems used. etc.) of which NEMO has very many and of course also on the forcing used to run the NEMO model.**

**Especially on the forcing side the HIDRA model has a great advantage in this comparison as it is allowed to use tide gauge data, whereas as I understand it the NEMO run they compare with does not assimilate sea level data. I would expect, although I don't know, that HIDRA3 without tide gauge data as inputs would perform worse that the specific NEMO run. Anyway, it should be made more clear in the text that although they outperform this specific Copernicus product it does not really imply much about the capabilities of numerical ocean models in general.**

RESPONSE 2:

Thank you for pointing this out. We now provide a more detailed description of the specific CMEMS Copernicus NEMO model setup in Section 3.1 and include a reference. We would like to clarify that the version of NEMO used in this paper incorporates sea-level data assimilation for satellite altimetry (SLA), but not for tide gauges. However, tide gauge measurements are used to correct the bias. This is discussed further in Response 6. Below is the new Section 3.1, which describes the version of NEMO used in this study:

**3.1  NEMO model description**

We compare HIDRA3 with the state-of-the-art deep model HIDRA2 (Rus et al., 2023) and with the standard Copernicus Marine Environment Monitoring Service (CMEMS) product MEDSEA_ANALYSISFORECAST_PHY_006_013 (Clementi et al., 2021) numerical model Nucleus for European Modelling of the Ocean (NEMO) v4.2 (Madec, 2016). The Mediterranean Sea Physical Analysis and Forecasting model (MEDSEA_ANALYSISFORECAST_PHY_006_013) operates on a regular grid with a horizontal resolution of 1/24° (approximately 4 km) and 141 vertical levels. It uses a staggered Arakawa C-grid with masking for land areas, and a z* vertical coordinate system with partial cells to accurately represent the model topography. The model incorporates tidal forcing using eight components (M2, S2, N2, K2, K1, O1, P1, Q1) and is forced at its boundaries by the Global analysis and forecast product (GLOBAL_ANALYSISFORECAST_PHY_001_024) on the Atlantic side, while in the Dardanelles Strait, it uses a combination of daily climatological fields from a Marmara Sea model and the global analysis product. Atmospheric forcing comes from ECMWF forecasting product. Initial conditions are taken from the World Ocean Atlas (WOA) 2013 V2 winter climatology as of January 1, 2015. Data assimilation is performed using the OceanVar (3DVAR) scheme, which integrates in-situ vertical profiles of temperature and salinity from ARGO, Glider, and XBT, as well as Sea Level Anomaly (SLA) data from multiple satellites (including Jason 2 & 3, Saral-Altika, Cryosat, Sentinel-3A/3B, Sentinel6A, and HY-2A/B). The hydrodynamic part of the model is coupled to the wave model WaveWatch-III. Further information about the model is available in Clementi et al. (2021).

Since NEMO predicts the full ocean state, including SSH, on a regular grid, the respective tide gauge locations are approximated by the nearest-neighbor locations in the grid. One important thing to note is that ocean models like NEMO calculate sea surface height as a local departure from the geoid in the computational cell. A typical cell dimension is of the order of hundreds of meters. This means that the model's SSH represents a departure from the geoid, averaged over hundreds of squared meters, and is not directly relatable to the in-situ observations from local tide gauges, which measure local water depth. Therefore, to align NEMO forecasts with local tide gauge water depth, an offset adjustment of the initial 12-hour forecast is necessary to ensure zero bias compared to the respective tide gauge, as explained in Rus et al. (2023).

**COMMENT 3:**

**2) The uncertainty quantifications and it's capabilities should be elaborated on more in the manuscript.**

**RESPONSE 3:**

We agree with the reviewer. To address this concern, we add a new Section 3.3, *"Uncertainty Estimation Analysis"*, to discuss and analyze HIDRA3's uncertainty prediction capabilities:

**3.3 Uncertainty estimation analysis**

Besides the forecasted SSH values, HIDRA3 also estimates uncertainties by predicting the standard deviation for each forecasted value. To assess the reliability of the estimated standard deviations, we employ the scaled error metric

$$\epsilon_{i,t} = \frac{\mu_{i,t} - \hat{\mu}_{i,t}}{\hat{\sigma}_{i,t}}, \tag{5}$$

as proposed by Barth et al. (2020). This metric quantifies the difference between the ground truth SSH value $\mu_{i,t}$ and the predicted SSH value $\hat{\mu}_{i,t}$, scaled by the estimated standard deviation $\hat{\sigma}_{i,t}$, where $i$ is the tide gauge index and $t$ is the prediction lead time. To assess the accuracy of our predicted uncertainty, we calculate the mean $\mu_\epsilon$ and the standard deviation $\sigma_\epsilon$ of the scaled error. If the predicted and the observed error distributions align, the standard deviation of the scaled error should be $\sigma_\epsilon = 1.0$.

For the second half of 2019 data, which was not used in training, the standard deviation of the scaled error for HIDRA3 is $\sigma_\epsilon = 0.98$. This indicates that HIDRA3 has good uncertainty prediction capabilities, with a slight overestimation of standard deviations of the prediction errors. To further analyze the distribution of the scaled error metric $\epsilon_{i,t}$, we plot its histogram in Fig. 13, along with the ideal model (a zero-centered unit-sigma Gaussian), and with the Gaussian distribution characterized by the estimated mean $\mu_\epsilon$ and standard deviation $\sigma_\epsilon$ of HIDRA3. Note that the estimated Gaussian distribution aligns well

with the distribution of the ideal uncertainty prediction model, suggesting that HIDRA3 has excellent uncertainty prediction capabilities.

[Figure]

**Figure 13.** Histogram of the scaled error $\epsilon_{i,t}$, overlaid with ideal Gaussian distribution with mean 0 and variance 1, and a Gaussian distribution characterized by the estimated mean $\mu_\epsilon$ and standard deviation $\sigma_\epsilon$ from HIDRA3. The estimated Gaussian distribution aligns well with the ideal one, suggesting excellent uncertainty prediction capabilities of HIDRA3.

**COMMENT 4:**

**3) The models architecture is well described, but the reasons for the modelling choices made is not. Perhaps, much of this information is available in earlier HIDRA papers, but I would like to see more motivations for the different modelling choices.**

**RESPONSE 4:**

We thank the reviewer for their suggestion to provide more explanations for our modelling choices. We include additional explanations and insights in Section 2.2.1, "*Geophysical encoder module*," and Section 2.2.2, "*Feature extraction module*," and revised Section 2.2.3, "*Feature fusion module*," and Section 2.2.4, "*SSH regression module*." Below, we include a latexdiff of Section 2.2.1 and 2.2.2, and the new Section 2.2.3 and Section 2.2.4.

[revised manuscript text omitted]

**COMMENT 5:**

**Specific comments:**

**L13 I think standard numerical model NEMO is the wrong label.**

**RESPONSE 5:**

We agree. We changed the label to be more specific into "the Mediterranean basin NEMO setup of the Copernicus CMEMS service."

**COMMENT 6:**

**L44 Goes back to general point 1), these comparisons are very much of apples and oranges**

**RESPONSE 6:**

The reviewer is raising an interesting point. We agree that it is often difficult to compare models between themselves or at least to attribute which level of performance corresponds to which algorithmic aspect of the model. In this sense, comparing ROMS to NEMO using different lateral boundary conditions, different atmospheric forcing and different parameterization schemes is no less difficult than comparing NEMO to HIDRA. We are unfortunately not in a position to feed the exact same atmospheric input and the exact same tide gauge input into NEMO (which does not ingest tide gauges as HIDRA3 but does assimilate satellite SLA) and into HIDRA3 (which does not ingest SLA as NEMO but receives tide gauges), therefore any comparison needs to keep in mind that the errors of both numerical approaches are accumulated across all input sources and the models themselves. Both NEMO and HIDRA3 are at the end of the modeling chain so their performance also reflects the accuracy of their input data.

We nevertheless feel that these comparisons can serve a purpose to establish the bottom line - *which of the models at our disposal gives the best forecast for civil rescue* and other emergency responses. In this optics, the comparisons presented in the paper are simply comparisons between the best sea level prediction setups at our disposal. We agree that these setups may be structurally quite heterogeneous but they all answer the same key question: what is the evolution of sea level in the next 72 hours? For the civil rescue response, coastal safety and the economy, this is a key issue.

We hope the reviewer agrees that the comparisons between admittedly heterogeneous modeling setups nevertheless hold some valuable information for the downstream services.

**COMMENT 7:**

**L193 The NEMO setup has to be better described. What NEMO version is used? What forcing is used (also temporal and spatial resolution). What is the vertical and horizontal resolution of the model. What vertical coordinate system is used? Does it have a wave model? Does it have data assimilation? Does it have a minimum depth? Information of that kind is needed to give more context to the different comparisons.**

**RESPONSE 7:**

We agree and have addressed this issue in our Response 2.

---

## Author Comment (AC2)

Dear editor, dear reviewer.

Before we proceed with point-by-point addressing of the issues raised by the reviewer, we would like to make a very short synopsis of what was done during this revision.

1. Based on the recommendation of Referee #2, we conducted experiments to test the failure of regional tide gauges. In light of this, we propose changing the title of the manuscript by removing the word "robust" and replacing it with a more descriptive phrase "in the presence of tide gauge sensor failures," so the new title reads: "HIDRA3: deep-learning model for multi-point ensemble sea level forecasting in the presence of tide gauge sensor failures."
2. We have added new sections to the manuscript: 2.4 "*Summary of differences to HIDRA2*," 3.1 "*NEMO model description*," 3.3 "*Uncertainty estimation analysis*," 3.4.2 "*Extreme scenario of a regional tide gauge failure*" and 3.7 "*HIDRA3 limitations*".
3. We have provided additional explanations and figures in the architecture description sections, rewriting Section 2.2.3, "*Feature fusion module*," and Section 2.2.4, "*SSH regression module*."
4. We have included explanations for the data quality check procedures and computation of the tidal signal, expanded the analysis of sea surface temperature and waves' impact on the performance of the model, and corrected some typos.

We proceed to a detailed point-by-point response below.

**COMMENT 1:**

**This study introduces HIDRA3, a deep-learning model developed to estimate multi-point sea levels. It builds upon previous work (Rus et al., 2023), aiming to enhance accuracy and handle missing data. While the research aligns with the scope of this journal and is generally well-written, several unclear aspects need to be addressed before the manuscript can be considered for publication.**

**RESPONSE 1:**

The authors thank the reviewer for their constructive remarks. We have done our best to amend the manuscript according to their suggestions. We respond point by point below.

**COMMENT 2:**

**Major Comments:**

**The major concern is the unclear methodology and application conditions for HIDRA3.**

**Page 5, Line 84: The phrase "detailed manual quality checks" is vague. Does this mean the authors removed data if outliers were detected? How were outliers defined? Is this process feasible in real time? If not, HIDRA3 has only been tested under ideal conditions where manual quality checks have already been applied, which may not be applicable in real-world scenarios. It is essential to clarify what data processing was conducted and if this is possible in real-time. If not, HIDRA3's performance should be tested on original data without quality checks.**

**RESPONSE 2:**

We agree with the reviewer that the data-cleaning process is important to the reader. Our data cleaning process addressed three types of errors, all of which could be automatically detected, making real-time implementation feasible. To enable automatic data cleaning, we defined thresholds based on our data-cleaning process and provide detailed descriptions of the data-cleaning steps in the manuscript:

In order to eliminate SSH measurement errors, we apply three filters to address three types of errors: (1) sensor freeze, which results in the same value being produced for an extended period of time; (2) extreme outliers; and (3) extreme jumps in the signals. For case (1), we define a threshold of 5 repetitions, as repetitions typically span more than 5 time points. For case (2), we define a location-specific threshold equal to 10 times the standard deviation of measurements and remove all points that are further away from the mean value than that threshold. Note that the threshold needs to be recalculated after each measurement is removed. For case (3), we examine the first derivative of the signals and define a location-specific threshold equal to 10 times the standard deviation of all first derivatives for some location. When two derivatives with opposite signs are close together (less than 10 time points), the region between them is removed. Again, it is necessary to recalculate the threshold after each removal. The validity of all thresholds was visually verified.

**COMMENT 3:**

**Page 5, Line 85: The manuscript suggests that tide data was predicted in one-year intervals. This may lead to "cheating" by using future data for tide predictions. For instance, predicting water levels on June 1, 2019, might involve tide data that includes water levels from that date. In real-world applications, future water level data would not be available. The authors should revise their approach to ensure that tide predictions are made only using available data, rather than yearly-based data.**

**RESPONSE 3:**

We thank the reviewer for pointing out this unclarity. We emphasize that the future data *was not used*. The tidal analyses used to compute tidal constituents during each particular year were computed from the past year of tide gauge observations except for the first year. The reason for performing tidal analysis in one-year chunks stems from the fact that tidal signals contain several low-frequency signals, most notably an 18.6-year cycle of the precession of lunar nodes. This leads to a slow oscillation in amplitudes and phases of tidal constituents. A classical approach to remedy this situation when computing tides from observations is to compute them in one-year chunks (see e.g. https://doi.org/10.1016/S0098-3004(02)00013-4). An alternative for longer time windows is to employ nodal corrections or perform analyses on series of sufficient length to resolve all necessary low frequency constituents. To address the reviewer's concern, we add the following text to the manuscript:

Astronomic tides were calculated from tide gauge data in one-year intervals using the UTIDE Tidal Analysis package for Python (Codiga, 2011). Using one year intervals for tidal analysis are common approach to compensate for the unresolved low frequency signals in the tidal signal, like the 18.6 year oscillation due to the precession of lunar nodes (Pawlowicz et al., 2002). For each one-year interval, tidal constituents are inferred from the past year of observations to ensure that no information beyond the prediction point is used. The only exception is in the first year of measurements for each location, when the first year of measurements is used to calculate the first year of tides. This approach is beneficial for training the model and does not affect the evaluation data, as all tide gauges have measurements spanning at least one year prior to the test period.

**COMMENT 4:**

**Page 5, Line 93: The reason for using ERA5 for training and ECMWF for testing is unclear. If the model trained with ERA5 outperforms the model trained with ECMWF, the authors should clearly present this result. Otherwise, the choice to use ERA5 for training is unjustified.**

**RESPONSE 4:**

We thank the reviewer for this comment. We have experimented with training using both ERA5 and ECMWF prediction data, and we discovered that training with ERA5 slightly enhances performance on the test set. This is why we decided to train with ERA5. We include a sentence in the manuscript to demonstrate this improvement:

ERA5 reanalysis data (Hersbach et al., 2018) was employed for training purposes, while ECMWF Ensemble Prediction System (EPS) data (Leutbecher and Palmer, 2007) was employed for evaluation to reflect the practical forecasting setup in which future reanalysis does not exist and forecasts are used. To independently verify the impact of the training set, HIDRA3 was also retrained on ECMWF EPS data instead of ERA5. When evaluated on the ECMWF EPS dataset, this led to a slight increase in MAE (1.7%). We consequently decided to select the ERA5 dataset for the training.

**COMMENT 5:**

**Page 7, Line 119: HIDRA3 incorporates additional features (sea surface temperature and waves) compared to HIDRA2, but this is not clearly stated. The manuscript should explain why these features were included. Although Section 3.4.2 discusses their impact, it does not analyze their individual contributions. The authors should reference feature selection studies and test the impact of each new feature (sea surface temperature and waves) to justify their inclusion.**

**RESPONSE 5:**

We thank the reviewer for their suggestion. We have added two ablation studies to Section 3.6.2 "*Impact of sea temperature and waves,*" in which we test the individual contributions of waves and sea surface temperature. From the experiments, it is difficult to conclusively state which feature contributes more, but it is nevertheless clear, that adding them slightly improves the performance. The revised Section 3.6.2, "*Impact of sea temperature and waves*," now reads as follows:

**3.6.2 Impact of sea temperature and waves**

To evaluate the importance of using sea surface temperature and wave data to enhance our model, we retrained HIDRA3 by these variables ablated. We first removed only sea surface temperature, which resulted in an average 1.7% increase in the overall MAE and a 1.2% increase in the MAE on high SSHs. Next, we removed only wave data, which led to an average 1.2% increase in the overall MAE and a 2.4% increase in the MAE on high SSHs. Finally, in the third experiment, we removed both sea surface temperature and wave data, which caused an average 0.8% increase in the overall MAE and a 2.7% increase in the MAE on high SSHs. These results suggest that sea surface temperature and wave data are not the main sources of the HIDRA3's excellent performance, however, they do bring a non-negligible contribution and we recommend using them in the SSH prediction tasks.

**COMMENT 6:**

**Sections 2.2.3 and 2.2.4: It is unclear how missing data (denoted as xi) is handled. Page 9, Line 150 mentions that missing values are estimated from "s," but it is unclear what "s" refers to. The authors need to clarify what the feature fusion module is doing and explain the difference between xi and s, preferably with a figure for better understanding.**

**RESPONSE 6:**

We thank the reviewer for this comment. We agree that the explanation of how missing values are handled is not clear. In the Feature Fusion module, location-specific features "xi" from only tide gauges with available measurements are combined into "s". We rewrite Section 2.2.3 "*Feature fusion module*" and include figures to clearly illustrate this process, as well as the reasons for passing "xi" to the SSH regression module.

**2.2.3 Feature fusion module**

As indicated in Fig. 4, the feature fusion module combines the location-specific features $x_i \in \mathbb{R}^{1024}$, into a joint state vector $s \in \mathbb{R}^{8192}$. A critical design requirement for the module is robustness to missing data; specifically, the number of location-specific features $x_i$ may vary, as they are only available for the tide gauges with available measurements.

Firstly, a partial reconstruction $s_i$ of the state is computed from each location-specific feature vector $x_i$ by applying a location-specific dense layer (see Fig. 8). In addition, a weight vector $w_i$ is computed by applying another location-specific dense layer to $x_i$. Each coordinate in $w_i$ reflects the extent by which the particular location contributes to the respective coordinate in the joint state vector. If some tide gauges are non-descriptive, their weights would be reduced during training, lowering their influence on the final state vector $s$, which is thus computed as

$$s = \sum_{i \in V} \hat{w}_i \odot s_i, \tag{1}$$

where $\odot$ denotes element-wise array multiplication, $\hat{w}_i$ are the coordinate-normalized weight vectors and $V$ contains indices of tide gauges with available SSH measurements, so that $s$ is approximated from tide gauges with available measurements. The components $\hat{w}_{i,j}$ are defined computed using softmax function:

$$\hat{w}_{i,j} = \frac{e^{w_{i,j}}}{\sum_{k \in V} e^{w_{k,j}}}, \tag{2}$$

where the softmax ensures that the coordinate-wise weights sum to one across all locations with available measurements.

[Figure]

**Figure 8.** The Feature Fusion Module takes features $\mathbf{x}_i$ from location $i$ and uses them to generate weights $\mathbf{w}_i$ and a partial reconstruction of state $\mathbf{s}_i$. These weights and partial reconstructions are combined into a joint state vector $\mathbf{s}$ using weighting and aggregation mechanisms. Locations without available measurements are excluded from the softmax calculation and aggregation. The parameters of the dense layers are specific to each location. The element-wise multiplication is denoted by the symbol $\odot$.

**COMMENT 7:**

**Page 11, Line 191: The differences between HIDRA2, NEMO, and HIDRA3 are not adequately explained. It would be helpful for the authors to provide a clear comparison of these models. For example, HIDRA2 does not consider temperature and wave data. Both HIDRA2 and HIDRA3 are designed for 72-hour predictions. NEMO, on the other hand, performs bias correction every 12 hours. Clarifying these differences would strengthen the manuscript.**

**RESPONSE 7:**

We thank the reviewer for pointing this out. We have provided a detailed description of NEMO in Section 3.1, titled "*NEMO model description,*" and outlined the differences between HIDRA2 and HIDRA3 in Section 2.4, "*Summary of differences to HIDRA2.*" Both are included below.

**2.4 Summary of differences to HIDRA2**

HIDRA3 presents a significant advancement over its predecessor, HIDRA2 (Rus et al., 2023), by introducing the capability to simultaneously process data from multiple tide gauge locations. This required a major redesign of the model, as HIDRA3 must effectively handle scenarios where SSH measurements are not available.

The only similar part is the Geophysical encoder module, but with a difference in the way temporal data is processed. In HIDRA2, there is a 4-hour temporal reduction of atmospheric data, while HIDRA3 incorporates the temporal reduction directly into the convolutional operations of the Geophysical encoder module. Additionally, HIDRA3 expands its input data to include sea surface temperature and wave fields, and considers not only the past 24 hours like HIDRA2, but 72 hours before the prediction point.

A notable contribution of HIDRA3 is its capacity for uncertainty quantification, a feature that was absent in HIDRA2. This capability is crucial for assessing the reliability and potential limitations of the SSH forecasts generated by the model.

**3.1 NEMO model description**

We compare HIDRA3 with the state-of-the-art deep model HIDRA2 (Rus et al., 2023) and with the standard Copernicus Marine Environment Monitoring Service (CMEMS) product MEDSEA_ANALYSISFORECAST_PHY_006_013 (Clementi et al., 2021) numerical model Nucleus for European Modelling of the Ocean (NEMO) v4.2 (Madec, 2016). The Mediterranean Sea Physical Analysis and Forecasting model (MEDSEA_ANALYSISFORECAST_PHY_006_013) operates on a regular grid with a horizontal resolution of 1/24° (approximately 4 km) and 141 vertical levels. It uses a staggered Arakawa C-grid with masking for land areas, and a z* vertical coordinate system with partial cells to accurately represent the model topography. The model incorporates tidal forcing using eight components (M2, S2, N2, K2, K1, O1, P1, Q1) and is forced at its boundaries by the Global analysis and forecast product (GLOBAL_ANALYSISFORECAST_PHY_001_024) on the Atlantic side, while in the Dardanelles Strait, it uses a combination of daily climatological fields from a Marmara Sea model and the global analysis product. Atmospheric forcing comes from ECMWF forecasting product. Initial conditions are taken from the World Ocean Atlas (WOA) 2013 V2 winter climatology as of January 1, 2015. Data assimilation is performed using the OceanVar (3DVAR) scheme, which integrates in-situ vertical profiles of temperature and salinity from ARGO, Glider, and XBT, as well as Sea Level Anomaly (SLA) data from multiple satellites (including Jason 2 & 3, Saral-Altika, Cryosat, Sentinel-3A/3B, Sentinel6A, and HY-2A/B). The hydrodynamic part of the model is coupled to the wave model WaveWatch-III. Further information about the model is available in Clementi et al. (2021).

Since NEMO predicts the full ocean state, including SSH, on a regular grid, the respective tide gauge locations are approximated by the nearest-neighbor locations in the grid. One important thing to note is that ocean models like NEMO calculate sea surface height as a local departure from the geoid in the computational cell. A typical cell dimension is of the order of hundreds of meters. This means that the model's SSH represents a departure from the geoid, averaged over hundreds of squared meters, and is not directly relatable to the in-situ observations from local tide gauges, which measure local water depth. Therefore, to align NEMO forecasts with local tide gauge water depth, an offset adjustment of the initial 12-hour forecast is necessary to ensure zero bias compared to the respective tide gauge, as explained in Rus et al. (2023).

COMMENT 8:

**Tables 2, 3, and 4: It is unclear why the recall, precision, and F1 scores for "low SSH values" are missing. Low water level predictions are important, particularly for critical infrastructure like nuclear power plants or harbors. The authors should explain why these metrics are missing and include them if possible.**

RESPONSE 8:

We agree with the reviewer that there is no reason not to include recall, precision, and F1 scores for low SSH values. We computed the metrics and added them to the tables:

| | Model | MAE (cm) | RMSE (cm) | ACC (%) | Bias (cm) | Re (%) | Pr (%) | F1 (%) |
|---|---|---|---|---|---|---|---|---|
| | NEMO | 2.65 | 3.56 | 97.76 | -0.31 | / | / | / |
| Overall | HIDRA2 | 2.63 | 3.56 | 98.15 | -0.17 | / | / | / |
| | HIDRA3 (ours) | **2.42** | **3.28** | **98.60** | **-0.00** | / | / | / |
| | NEMO | 4.19 | 5.23 | 92.91 | 2.88 | 94.04 | **99.92** | 96.39 |
| Low SSH Values | HIDRA2 | **3.27** | 4.27 | 95.94 | **1.02** | 97.64 | 99.55 | 98.51 |
| | HIDRA3 (ours) | 3.30 | **4.24** | **96.16** | 1.33 | **98.04** | **99.85** | **98.88** |
| | NEMO | 4.68 | 6.19 | 89.14 | -3.02 | 94.53 | **99.40** | 96.79 |
| High SSH Values | HIDRA2 | 4.80 | 6.53 | 89.49 | -2.35 | 96.62 | 97.82 | 97.18 |
| | HIDRA3 (ours) | **4.06** | **5.61** | **91.63** | **-2.06** | **97.58** | 98.67 | **98.09** |

**Table 2.** Performance calculated on all SSH values, low SSH values and high SSH values, averaged over all locations. The proposed HIDRA3 has the best performance overall and on high SSH values, and a comparable performance on low values to HIDRA2.

| | Model | MAE (cm) | RMSE (cm) | ACC (%) | Bias (cm) | Re (%) | Pr (%) | F1 (%) |
|---|---|---|---|---|---|---|---|---|
| Overall | $NEMO_0$ | 3.26 | 4.15 | 95.81 | **0.03** | / | / | / |
| | HIDRA3 (ours) | **2.63** | **3.52** | **98.35** | -0.07 | / | / | / |
| Low SSH | $NEMO_0$ | 4.00 | 4.95 | **96.00** | 3.08 | 97.41 | **99.55** | 98.44 |
| Values | HIDRA3 (ours) | **3.30** | **4.26** | 95.75 | **1.02** | **98.23** | 99.51 | **98.82** |
| High SSH | $NEMO_0$ | 5.12 | 6.48 | 86.82 | -2.96 | 92.20 | **99.81** | 95.24 |
| Values | HIDRA3 (ours) | **4.46** | **6.04** | **89.94** | **-2.32** | **97.38** | 98.81 | **98.06** |

**Table 3.** Performance of HIDRA3 and $NEMO_0$ under the target location tide gauge failure.

| | Model | MAE (cm) | RMSE (cm) | ACC (%) | Bias (cm) | Re (%) | Pr (%) | F1 (%) |
|---|---|---|---|---|---|---|---|---|
| | NEMO | 2.65 | 3.56 | 97.76 | -0.31 | / | / | / |
| Overall | HIDRA2 | 2.63 | 3.56 | 98.15 | -0.17 | / | / | / |
| | $HIDRA3_1$ (ours) | **2.60** | **3.47** | **98.40** | 0.02 | / | / | / |
| | NEMO | 4.19 | 5.23 | 92.91 | 2.88 | 94.04 | **99.92** | 96.39 |
| Low SSH Values | HIDRA2 | **3.27** | **4.27** | **95.94** | **1.02** | **97.64** | 99.55 | **98.51** |
| | $HIDRA3_1$ (ours) | 3.52 | 4.47 | 95.58 | 1.79 | 97.31 | 99.21 | 98.16 |
| | NEMO | 4.68 | 6.19 | 89.14 | -3.02 | 94.53 | **99.40** | 96.79 |
| High SSH Values | HIDRA2 | 4.80 | 6.53 | 89.49 | -2.35 | 96.62 | 97.82 | 97.18 |
| | $HIDRA3_1$ (ours) | **4.34** | **5.98** | **90.94** | **-1.72** | **96.82** | 98.54 | **97.65** |

**Table 4.** Performance of NEMO, HIDRA2 and $HIDRA3_1$, where $HIDRA3_1$ is the model trained separately on every single location.

**COMMENT 9:**

**Figure 11: The manuscript only considers "pair" failures for tide gauges, but more realistic scenarios should be explored. For instance, a failure involving multiple tide stations, such as in the northern Adriatic (KP, VE, RA), where water levels show similar trends (as per Figure 2), would offer a more realistic test of HIDRA3's performance. Testing such scenarios would strengthen the justification for using HIDRA3 over other approaches.**

**RESPONSE 9:**

We thank the reviewer for this excellent experiment, in which we simulated the complete failure of all tide gauges in Koper, Venice, and Ravenna. The results of the experiment demonstrate that HIDRA3 remains robust in the case of regional tide gauge failure, with only a slight downgrade in performance observed for the northern locations due to the removed measurements. The results of this experiment are detailed in Section 3.4.2 of the revised manuscript, titled *"Extreme scenario of a regional tide gauge failure:"*

**3.4.2 Extreme scenario of a regional tide gauge failure**

To evaluate the model's performance under an extreme scenario of a regional tide gauge failure, we conducted an additional experiment where all northern locations (Koper, Venice, and Ravenna) were disabled. These locations exhibit distinct dynamical characteristics compared to others, as illustrated in Figures 2 and 3. The evaluation of HIDRA3 without SSH measurements from Koper, Venice, and Ravenna is denoted as HIDRA3$_S$. Importantly, the model was not retrained for this experiment; rather, the same model trained on all locations was used, with the specified measurements withheld from the test set. This stress test quantifies HIDRA3 predictions for northern tide gauges based exclusively on data from southern tide gauges.

Figure 17 compares the MAE scores of HIDRA3$_S$, HIDRA3, and NEMO. Surprisingly, HIDRA3$_S$ demonstrates robust forecasting capabilities in the northern locations, despite lacking direct measurements. While the MAE scores are slightly higher than those obtained with all locations as inputs (HIDRA3), they significantly outperform NEMO. In the southern locations, the performance of HIDRA3 and HIDRA3$_S$ is comparable and worse than NEMO. This is expected given the availability of past SSH measurements for these regions.

We plot the performance during multiple tide gauge failure in a separate Figure:

[Figure]

**Figure 17.** Multiple sensor failure scenario. Overall (dark) and high SSH (light) MAE scores for NEMO, HIDRA3, and HIDRA3$_S$. HIDRA3$_S$ represents the evaluation of HIDRA3 with only southern locations as input, excluding sensors in Koper, Venice, and Ravenna.

We also add a sentence to the Abstract:

respectively, setting a solid new state-of-the-art. Forecasting skill does not deteriorate even in the case of simultaneous failure of multiple sensors in the basin or when predicting solely from the tide gauges far outside the Rossby radius of a failed sensor. Furthermore, HIDRA3 shows remarkable performance at substantially smaller amounts of training data compared with HIDRA2, making it appropriate for sea level forecasting in basins with large regional variability in the available tide gauge data.

**COMMENT 10:**

**There is no dedicated section on the limitations of HIDRA3. For example, HIDRA3 does not work if data from at least one station is missing for the 72-hour prediction window. The limitations should be clearly stated.**

**RESPONSE 10:**

To ensure a comprehensive understanding of HIDRA3's capabilities and limitations, we have added a dedicated section to our paper that outlines the specific conditions under which the model may encounter challenges:

**3.7  HIDRA3 limitations**

HIDRA3 can generate forecasts for all locations even if the data from some locations is missing, which is a substantial improvement over its predecessor, HIDRA2. The current implementation of HIDRA3 disregards the entire 72-hour SSH dataset from the previous three days if even a single value of SSH is missing. This might be an overly stringent criterion and we are working to find optimal ways to address this. Furthermore, in a highly unlikely situation when the sensors at *all locations* would fail and the data would be missing for the full 72 h past period, HIDRA3 could not generate forecasts reliably. In this case, a potential solution of using the past HIDRA3 prediction as the input for the current prediction is still applicable. Preliminary simulations of such scenarios indicate that the MAE of HIDRA3 increases by 35% overall and by 43% for high SSH values. Additional work, however, needs to be performed to further constrain HIDRA3 behavior in such scenarios.

**COMMENT 11:**

**Minor Comments:**

**Page 3, Line 45: The full name of HIDRA should be provided.**

**RESPONSE 11:**

Thank you. The text was corrected as suggested.

**COMMENT 12:**

**Page 3, Line 60: The authors should expand their literature review to include studies that address missing data in real time, such as Lee and Park (2016) and Vieira et al. (2020), for better context.**

**RESPONSE 12:**

We thank the reviewer for suggesting the relevant articles by Lee and Park (2016) and Vieira et al. (2020). We expand our literature review to include these contributions.

**COMMENT 13:**

**Page 5, Line 86-89: It would be helpful to clarify where the "high" and "low" data will be used in the next section. As written, the reason for defining "high" and "low" is unclear.**

**RESPONSE 13:**

We thank the reviewer for this suggestion. We now explicitly mention that we use these data in evaluation in the appropriate Section and we also list why this is relevant to know. The new paragraph now reads:

In the context of this analysis, a dataset of sea-level extremes was constructed for each station. SSH readings are categorized as *low* if they are below the $1^{st}$ percentile, and as *high* if they surpass the $99^{th}$ percentile of the observed values at the respective location. During evaluation (Sect. 3.2), metrics are calculated for all SSH values, as well as for low and high values. This helps assess the model's performance in predicting both tails of extreme SSH values: high values are relevant for coastal flood warnings while low values restrict marine traffic in the shallow north of the basin. The thresholds determined are listed in Table 1.

COMMENT 14:

**Figure 2 and Page 5, Line 89: The location names in Figure 2 and the acronyms used in the text should be consistent for readability.**

RESPONSE 14:

As the reviewer suggested, we updated the text to include the full names of the locations.

COMMENT 15:

**Figure 5 and Page 7, Line 118: The output dimensions are different for various features (e.g., wind and pressure have different dimensions compared to others). The caption for Figure 5 should be corrected to reflect these differences.**

RESPONSE 15:

We thank the reviewer for noticing this, we changed the caption as follows:

**Figure 5.** The first step of the Geophysical encoder module involves encoding each of the four variables separately. Note that wind has two input channels, while the waves data has four. The encoder consists of two 3D convolutions, reducing spatial dimension to $1 \times 1$. Wind and pressure are encoded into 512 output channels, as depicted in the figure, while sea surface temperature and wave data are encoded into 64 channels. The variables $k$, $s$ and $n$ denote the kernel size, the stride and the number of output channels, respectively. The number of channels is indicated in gray, the size of the temporal dimension is in red, and the spatial dimensions are in blue.

COMMENT 16:

**Figure 6: The authors should explain why the input dimension is 1152*36, given that the output dimension of Figure 5 is 512*36*1*1. The change in dimensions needs clarification.**

RESPONSE 16:

To address the reviewer's concern, we add the following explanation to the revised manuscript:

Encodings from all variables are concatenated, resulting in a total of $2 \cdot 512 + 2 \cdot 64 = 1152$ channels. By removing spatial dimensions of size 1, concatenation is of size $1152 \times 36$, which is then in the second step of the Geophysical encoder (see Fig. 6) processed by a 1D convolution with 256 kernels of temporal size 5. We have chosen the kernel size of 5 to increase

COMMENT 17:

**Figures 6 and 7: The terms "2X" and "4X" should be explained, as their meaning is unclear.**

RESPONSE 17:

As suggested, we added the explanations.

COMMENT 18:

**Page 8, Line 130: The authors mention a "dense layer," but later refer to "dropout" in Figure 7. This needs clarification, as dropout is not typically associated with fully connected layers.**

RESPONSE 18:

After the first dense layer mentioned in Line 130, we do not apply Dropout. However, after dense layers with residual connections, we do apply Dropout. We add dropout to prevent overfitting, and we have added this information to the manuscript. We hope we have addressed the reviewers' concerns.

COMMENT 19:

**Section 2.2.3 and 2.2.4: Including a diagram, similar to Figure 6, would help readers understand the concepts better.**

RESPONSE 19:

AS suggested, we added the following diagrams:

[Figure]

**Figure 8.** The Feature Fusion Module takes features $\mathbf{x}_i$ from location $i$ and uses them to generate weights $\mathbf{w}_i$ and a partial reconstruction of state $\mathbf{s}_i$. These weights and partial reconstructions are combined into a joint state vector $\mathbf{s}$ using weighting and aggregation mechanisms. Locations without available measurements are excluded from the softmax calculation and aggregation. The parameters of the dense layers are specific to each location. The element-wise multiplication is denoted by the symbol $\odot$.

[Figure]

**Figure 9.** The SSH regression module for a location $i$ receives joint state vector $\mathbf{s}$ and location-specific features $\mathbf{x}_i$. The features $\mathbf{x}_i$ are processed by a dense layer to produce the features $\hat{\mathbf{x}}_i$. In cases where measurements from the tide gauge at location $i$ are unavailable, and therefore $\mathbf{x}_i$ is undefined, $\hat{\mathbf{x}}_i$ is approximated from $\mathbf{s}$ using a separate dense layer. The features $\hat{\mathbf{x}}_i$ and $\mathbf{s}$ are then concatenated and passed through a final dense layer to obtain SSH predictions, denoted as $\hat{\boldsymbol{\mu}}_i$.

**COMMENT 20:**

**Page 9, Line 158: The term "mean" in "SSH mean value prediction" is unclear. The authors should clarify whether this is a typo or explain its meaning.**

**RESPONSE 20:**

We thank the reviewer for spotting this. It is a typo and we corrected it in the revision.

**COMMENT 21:**

**Page 10, Line 163: The standardization process needs further explanation. Was the data normalized for each case or across the entire dataset?**

**RESPONSE 21:**

The geophysical variables are standardized independently, while the tide gauges share the standard deviation used for their normalization. We add the following text in the revised manuscript to reflect this:

probability of 0.5. The batch size is set to 128 data samples, the model is trained for 20 epochs. All input data is standardized by subtracting the mean and dividing by the standard deviation. For each tide gauge location, the mean is calculated independently, while a single standard deviation is computed across all locations. Each geophysical variable is standardized independently of the other variables. Training takes approximately one hour on a computer with an NVIDIA A100 Tensor Core GPU graphics card.

COMMENT 22:

**Page 10, Line 170: For consistency, the term "first stage" should be replaced with "first phase."**

RESPONSE 22:

Thank you for your suggestion. We replaced "first stage" with "first phase."

COMMENT 23:

**Page 11, Line 193: The full name of NEMO should be provided.**

RESPONSE 23:

As suggested, we added the name: **"**Nucleus for European Modeling of the Ocean."

---

## Author Response (AR2)

Dear editor, dear reviewers.

We appreciate your encouraging comments and thorough review of our work. Below is our point-by-point response to the second report, along with our revisions based on the reviewer's comments.

**COMMENT 1:**

**The authors have thoroughly revised the manuscript, significantly enhancing its clarity. However, one comment related to Table 1 remains, originating from an unclear description in the original manuscript. Additionally, I have three suggestions connected to this comment.**

**1. For Comments 6 and 10: In my initial review, I misunderstood that HIDRA3 could handle a 72-hour time series with some missing values by filling the gaps. For this reason, I also thought the SSH availability reported in Table 1 represented the total number of scenarios used to train and evaluate the model. However, after reviewing the authors' response and the revised manuscript, it appears that HIDRA3 excludes any tide station data if there is even one missing data point in the 72-hour series. Consequently, the SSH availability in Table 1 may overstate the actual number of usable scenarios, as only a subset of cases contains a complete 72-hour time series. If this interpretation is correct, I recommend the authors clarify the exact sample size used for model training and testing to ensure readers understand the number of scenarios ultimately included.**

**RESPONSE 1:**

We agree with the reviewer that Table 1 should reflect the sample sizes used for training and testing. We updated the table with the new SSH availability. For stations with SSH signals without interruptions, the numbers remain unchanged.

The new table (not latex-diff) and the text changes (latex-diff) are displayed below:

**2.1 HIDRA3 training and testing datasets**

Our objective is to forecast hourly SSH values for $N = 11$ tide gauges located along the Adriatic coast (Fig. 1) over a three-day period. HIDRA3 achieves this by leveraging a comprehensive set of ocean state parameters. This includes the past 72 hours of available sea level observations from stations shown in Fig. 1, with data availability for each station detailed in Table 1.  When calculating the availability, only SSH measurements with 72 preceding measurements available are considered, as required for HIDRA3 input. Besides past SSH measurements, HIDRA3 considers both past and future astronomic tides at  the stations, and past and future 72 hours of gridded geophysical variables from atmospheric and ocean numerical models.

| Location | SSH Availability in 2000–2022 | Thresholds [cm] |
|---|---|---|
| Koper | 90.8% | -69.3, 65.7 |
| Venice | 64.6% | -64.3, 61.3 |
| Ancona | 50.4% | -39.9, 44.6 |
| Ortona | 45.3% | -34.0, 39.6 |
| Vieste | 44.9% | -33.3, 36.5 |
| Neretva | 38.9% | -32.6, 37.8 |
| Ravenna | 37.7% | -56.3, 57.2 |
| Sobra | 24.1% | -33.4, 37.0 |
| Stari Grad | 23.9% | -34.0, 38.7 |
| Tremiti | 18.2% | -32.4, 37.0 |
| Vela Luka | 16.6% | -31.9, 38.6 |

**Table 1.** Availability of SSH measurements between 2000 and 2022 for 11 tide gauge locations used in training and evaluating HIDRA3, and defined thresholds [1st, 99th percentile] for low and high SSH values used in this study. When calculating SSH Availability, only SSH measurements with 72 preceding measurements available are considered, as required for HIDRA3 input. See Fig. 1 for station locations.

**COMMENT 2:**

**2. Although these additional minor comments were not raised in the first review due to misunderstanding the SSH data and Table 1, I believe they would help improve the manuscript.**

**2.1. For Figure 3, how was this plot created if each station had a different amount of available data? Was only the overlapping data for all stations used? If so, please clarify whether this approach is valid.**

**RESPONSE 2:**

We used overlapping data for each pair of locations individually. These differences, shown in Figure 3, estimate the increase in MAE if we were to forecast SSH at location A with some model and then apply that forecast to location B without any modifications. While the amount of data used varies for each pair, our primary goal is to accurately estimate the increase in MAE. Therefore, we decided to utilize as much data as possible, prioritizing the correctness of the scores themselves rather than the comparability of different scores. We have revised the manuscript as follows:

of each other and thus exhibit similar SSH phases of high and low sea levels. This is illustrated in Fig. 2, which depicts the SSH at all stations, and in Fig. 3, which shows mean absolute differences between all stations. These mean absolute differences were calculated using overlapping data from each pair of locations, and they can be interpreted as estimates of the increase in mean absolute error (MAE) when applying some model's forecast from one location to another.

**Figure 3.** Mean absolute differences [cm] of SSH measurements between different tide gauge locations. These differences estimate the increase in MAE when applying some model's forecast from one location to another. Abbreviations used here are: KP - Koper, VE - Venice, RA - Ravenna, AN - Ancona, OR - Ortona, TR - Tremiti, VI - Vieste, SO - Sobra, VL - Vela Luka, NE - Neretva, and SG - Stari Grad.

**COMMENT 3:**

**2.2. Figures 10, 14, 17, and 19: Based on Figures 2 and 3, it appears that Koper, Venice, and Ravenna have larger MAE values than other stations due to their wider water level ranges. Additionally, each station may have a different number of test scenarios based on the SSH availability reported in Table 1. This variation raises questions about whether the MAE accurately reflects performance at each station. I suggest presenting normalized error statistics instead. Please also clarify if different sample sizes were used to calculate the error statistics and if comparing these across stations with different sample sizes is valid.**

**RESPONSE 3:**

We agree with the reviewer that the higher MAE scores observed in northern locations are related to greater variances in SSH in those regions. To expose this, we calculated the normalized MAE (nMAE) scores and presented them in Fig. 11. This plot provides additional insights: even after normalization, NEMO shows the highest errors in northern locations. In contrast, HIDRA2 exhibits the highest normalized errors in southern locations, likely due to lower data availability in those areas. We have also calculated the mean nMAE scores and included them in Tables 2, 3, and 4.

However, it is still important to include MAE scores without normalization. Potential users of HIDRA tend to be more interested in MAE scores expressed without normalization, as these scores have immediate practical significance. Additionally, MAE without normalization more effectively reflects flood forecasting capability, since higher SSH variability indicates a greater likelihood of flooding.

Additions made to the manuscript:

**3.2 SSH forecast performance**

The following performance measures (Rus et al., 2023) are employed: mean absolute error (MAE), root mean squared error (RMSE), accuracy (ACC), bias, recall (Re), precision (Pr) and F1 score. Additionally, we calculate the normalized mean absolute error (nMAE) by dividing the MAE score for each location by the standard deviation of all historic SSH measurements for that location. These performance metrics are reported in Table 2 separately for all SSH values (*overall*) and for low and high SSH values (see Sect. 2 for the definitions).

| | Model | MAE (cm) | nMAE | RMSE (cm) | ACC (%) | Bias (cm) | Re (%) | Pr (%) | F1 (%) |
|---|---|---|---|---|---|---|---|---|---|
| Overall | NEMO | 2.65 | 0.142 | 3.56 | 97.76 | -0.31 | / | / | / |
| | HIDRA2 | 2.63 | 0.146 | 3.56 | 98.15 | -0.17 | / | / | / |
| | HIDRA3 (ours) | **2.42** | **0.134** | **3.28** | **98.60** | **-0.00** | / | / | / |
| Low SSH Values | NEMO | 4.19 | 0.215 | 5.23 | 92.91 | 2.88 | 94.04 | **99.92** | 96.39 |
| | HIDRA2 | **3.27** | **0.175** | 4.27 | 95.94 | **1.02** | 97.64 | 99.55 | 98.51 |
| | HIDRA3 (ours) | 3.30 | 0.177 | **4.24** | **96.16** | 1.33 | **98.04** | 99.85 | **98.88** |
| High SSH Values | NEMO | 4.68 | 0.244 | 6.19 | 89.14 | -3.02 | 94.53 | **99.40** | 96.79 |
| | HIDRA2 | 4.80 | 0.266 | 6.53 | 89.49 | -2.35 | 96.62 | 97.82 | 97.18 |
| | HIDRA3 (ours) | **4.06** | **0.220** | **5.61** | **91.63** | **-2.06** | **97.58** | 98.67 | **98.09** |

**Table 2.** Performance calculated on all SSH values, low SSH values and high SSH values, averaged over all locations. The proposed HIDRA3 has the best performance overall and on high SSH values, and a comparable performance on low values to HIDRA2.

[Figure]

**Figure 11.** The normalized MAE (nMAE) calculated for all SSH values and for high SSH values, across different models and tide gauge locations. HIDRA3 demonstrated the most consistent performance, significantly outperforming NEMO (Madec, 2016) at northern locations (Koper, Venice and Ravenna), and HIDRA2 (Rus et al., 2023) at other locations.

To enable a more effective comparison of errors across different locations, we present the normalized MAE scores (nMAE) in Fig. 11. Although the scores are normalized, NEMO still shows the largest errors at northern locations (Koper, Venice and Ravenna). In contrast, HIDRA2 records larger normalized errors at the southern locations, likely due to lower data availability in those areas (see Table 1). HIDRA3 demonstrates the most consistent performance, significantly outperforming NEMO in the northern locations and HIDRA2 in the southern locations. On average, HIDRA3 has a lower nMAE score than both NEMO and HIDRA2 when calculated on all SSH values and high SSH values (see Table 2).

|  | Model | MAE (cm) | nMAE | RMSE (cm) | ACC (%) | Bias (cm) | Re (%) | Pr (%) | F1 (%) |
|---|---|---|---|---|---|---|---|---|---|
| Overall | $NEMO_0$ | 3.26 | 0.173 | 4.15 | 95.81 | **0.03** | / | / | / |
|  | HIDRA3 (ours) | **2.63** | **0.146** | **3.52** | **98.35** | -0.07 | / | / | / |
| Low SSH | $NEMO_0$ | 4.00 | 0.217 | 4.95 | **96.00** | 3.08 | 97.41 | **99.55** | 98.44 |
| Values | HIDRA3 (ours) | **3.30** | **0.176** | **4.26** | 95.75 | **1.02** | **98.23** | 99.51 | **98.82** |
| High SSH | $NEMO_0$ | 5.12 | 0.255 | 6.48 | 86.82 | -2.96 | 92.20 | **99.81** | 95.24 |
| Values | HIDRA3 (ours) | **4.46** | **0.245** | **6.04** | **89.94** | **-2.32** | **97.38** | 98.81 | **98.06** |

**Table 3.** Performance of HIDRA3 and $NEMO_0$ under the target location tide gauge failure.

|  | Model | MAE (cm) | nMAE | RMSE (cm) | ACC (%) | Bias (cm) | Re (%) | Pr (%) | F1 (%) |
|---|---|---|---|---|---|---|---|---|---|
| Overall | NEMO | 2.65 | **0.142** | 3.56 | 97.76 | -0.31 | / | / | / |
|  | HIDRA2 | 2.63 | 0.146 | 3.56 | 98.15 | -0.17 | / | / | / |
|  | $HIDRA3_1$ (ours) | **2.60** | 0.144 | **3.47** | **98.40** | **0.02** | / | / | / |
| Low SSH | NEMO | 4.19 | 0.215 | 5.23 | 92.91 | 2.88 | 94.04 | **99.92** | 96.39 |
| Values | HIDRA2 | **3.27** | **0.175** | **4.27** | **95.94** | **1.02** | **97.64** | 99.55 | **98.51** |
|  | $HIDRA3_1$ (ours) | 3.52 | 0.190 | 4.47 | 95.58 | 1.79 | 97.31 | 99.21 | 98.16 |
| High SSH | NEMO | 4.68 | 0.244 | 6.19 | 89.14 | -3.02 | 94.53 | **99.40** | 96.79 |
| Values | HIDRA2 | 4.80 | 0.266 | 6.53 | 89.49 | -2.35 | 96.62 | 97.82 | 97.18 |
|  | $HIDRA3_1$ (ours) | **4.34** | **0.239** | **5.98** | **90.94** | **-1.72** | **96.82** | 98.54 | **97.65** |

**Table 4.** Performance of NEMO, HIDRA2 and $HIDRA3_1$, where $HIDRA3_1$ is the model trained separately on every single location.

Different sample sizes were used to calculate the error statistics, and we agree with the reviewer that this diminishes the validity of comparing the errors between stations. However, we prefer not to calculate the errors only at the time points where SSH data is available for all locations, as this would decrease the number of samples used in computing metrics at certain locations, undermining the validity of model comparisons at those sites. Fortunately, nearby stations tend to have similar data availability (as detailed below), so we have decided to retain the analysis in the manuscript as it is.

SSH data availability in the test period:

```
Koper
 2019: 100.0%
 2020: 100.0%

Venice
 2019: 99.7%
 2020: 93.0%

Ravenna
 2019: 0.0%
 2020: 97.1%

Ancona
 2019: 96.7%
 2020: 99.8%

Ortona
 2019: 0.0%
 2020: 99.1%

Tremiti
 2019: 0.0%
 2020: 92.8%

Vieste
 2019: 0.0%
 2020: 99.2%

Neretva
 2019: 99.9%
 2020: 100.0%

Sobra
 2019: 100.0%
 2020: 99.9%

Stari Grad
 2019: 99.9%
 2020: 99.4%

Vela Luka
 2019: 91.3%
 2020: 99.8%
```

**COMMENT 4:**

**2.3. In Section 2.1 (line 84), the period from January 2019 to June 2019 is omitted from both training and testing. Is there a specific reason for this gap?**

**RESPONSE 4:**

At the time of development CMEMS NEMO forecasting products were available only after June 2019, which is why we do not include the first part of 2019 in our evaluation. However, we should use that data for training to extend the training period.